# Sensory stimuli dominate over rhythmic electrical stimulation in modulating behavior

**Yuranny Cabral-Calderin** [1]*, **Molly J. Henry** [1,2]*

**1** Max Planck Institute for Empirical Aesthetics, Frankfurt am Main, Germany, **2** Toronto Metropolitan University, Toronto, Canada

* yuranny.cabral-calderin@ae.mpg.de (YC-C); molly.henry@ae.mpg.de (MJH)

## Abstract

Neural tracking (entrainment) of auditory rhythms enhances perception. We previously demonstrated that transcranial alternating current stimulation (tACS) can enhance or suppress entrainment to rhythmic auditory stimuli, depending on the timing between the electrical and auditory signals, although tACS effects are primarily modulatory. This study further investigated entrainment to tACS and auditory rhythms when the electrical and auditory signals were presented together (Experiment 1, $N = 34$) or independently (Experiment 2, $N = 24$; Experiment 3, $N = 12$). We hypothesized that tACS effects would be more pronounced when the auditory rhythm was made less perceptually salient to reduce the competition with the electrical rhythm. Participants detected silent gaps in modulated or unmodulated noise stimuli. In Experiment 1, auditory stimuli predominated in entraining behavior. While behavioral entrainment to sound rhythms was affected by the modulation depth of the auditory stimulus, entrainment to tACS was not. In Experiment 2, with no rhythmic information from the sound, 17 of 24 participants showed significant behavioral entrainment to tACS, although the most effective tACS frequency varied across participants. An oscillator model with a free parameter for the individual resonance frequency produced profiles similar to those we observed behaviorally. In Experiment 3, both neural and behavioral entrainment to rhythmic sounds were affected by the auditory stimulus frequency, but again the most effective entraining frequency varied across participants. Our findings suggest that tACS effects depend on the individual's preferred frequency when there is no competition with sensory stimuli, emphasizing the importance of targeting individual frequencies in tACS experiments. When both sensory and electrical stimuli are rhythmic and compete, sensory stimuli prevail, indicating the superiority of sensory stimulation in modulating behavior.

provided the original author and source are credited.

**Data availability statement:** All data needed to evaluate the conclusions in the paper are present in the paper and/or the Supplementary Materials. Numerical data included in the figures have been made publicly available in https://doi.org/10.12751/g-node.98i4rq. The raw data can be shared upon request. All custom code used in the manuscript can be found in https://doi.org/10.5281/zenodo.15230985.

**Funding:** This work was supported by a European Research Council Starting Grant (BRAINSYNC, https://erc.europa.eu/apply-grant/starting-grant) and a Max Planck Research Group (https://www.mpg.de/en) granted to MJH. The funding institutions did not play any role in the study design, data collection and analysis, decision to publish, or preparation of the manuscript.

**Competing interests:** The authors have declared that no competing interests exist.

**Abbreviations:** AIC, Akaike's information criteria; AUC, area under the curve; FA, false alarms; FFT, fast Fourier transform; FMfrequency modulationrms, root mean square; ROC, receiver operating characteristic curve; rANOVA, repeated measures analysis of variance; tACS, transcranial alternating current stimulation; VL, vector length.

## Introduction

Rhythmic acoustic patterns are commonly found in spoken language, musical compositions, and environmental sounds. Increasing evidence suggests that neural activity can track (quasi)periodic stimulus information via neural entrainment [1–4]. Neural entrainment, characterized by the synchronization of neural signals with rhythmic auditory stimuli, concentrates phases of high neural excitability at moments when significant auditory events are anticipated, thereby increasing the probability that the stimulus elicits a neural response and reaches awareness. Consequently, neural entrainment may serve as a potential mechanism for modulating neural sensory gain [1,5] and is believed to enhance the perception and comprehension of auditory sensory information [3,4,6–12].

Neural entrainment extends beyond the human auditory system and is also present in other sensory modalities [13,14] as well as in various non-human species [15,16]. Notably, brain rhythms synchronize not only with rhythmic sensory stimulation but also with imposed external electric fields. Transcranial alternating current stimulation (tACS) is a commonly used method for noninvasively entraining brain rhythms in humans. This technique involves administering a low-intensity alternating current through scalp electrodes, with the goal of synchronizing neuronal activity to the electric input [17–20]. In the auditory domain, tACS has been used to influence speech comprehension, stream segregation, and binaural integration [21–28]. Interestingly, in a cocktail-party scenario, speech comprehension can be modulated by tACS when the stimulation is synchronized with the speech envelope of attended and ignored speech signals [29], supporting the efficacy of tACS in modulating stimulus–brain synchrony and auditory performance.

Despite the capacity of brain rhythms to align with rhythmic stimuli or external electrical stimulation, the details of the interaction or competition between auditory and electrical entraining rhythms remains unclear. This is important because tACS effects are often weak, and without quantifying the relative contributions of auditory and electrical rhythms to entrainment, we cannot know how strongly tACS can push brain rhythms around while it is competing with rhythmic sensory stimulation. We do not assume entrainment to electrical and auditory rhythms to be independent; significant interactions, such as ceiling effects and other nonlinear dynamics, are likely to occur [30]. In our recent research, we combined frequency-modulated (FM) sounds with concurrent tACS [31]. The findings indicated that tACS could enhance or diminish entrainment to auditory stimulation, depending on the phase relationship between auditory and electrical signals. However, auditory stimuli played a dominant role in guiding behavior on a trial-by-trial basis. We interpreted this result as the superiority of the sensory signal in entraining behavior [31]. That is, when both the acoustic signal and tACS compete to entrain behavior, tACS can only modulate the strength of behavioral entrainment to the sound but cannot overcome the significant behavioral modulation driven by the acoustic signal.

In the current study, we delved deeper into this question by investigating entrainment to tACS and auditory rhythms across three experiments. In the first experiment,

we applied tACS while participants listened to FM sounds modulated at 2 Hz, but with two different modulation depths. We asked whether tACS effects depend on the perceptual salience of the auditory rhythm under the hypothesis that tACS effects should be strongest when the auditory stimulus is weakly modulated. In the second experiment, we applied tACS at different frequencies while participants listened to unmodulated noise stimuli and asked whether tACS could modulate perception on its own when no rhythmic information was conveyed by the sound. In addition, we used a computational model to explore the role of individual resonance frequencies of hypothetical oscillators on tACS effects. In the third and final experiment, we collected EEG data while participants listened to FM sounds at different frequencies, exploring individual variability in neural and behavioral entrainment to acoustic signals in the absence of electrical stimulation and relating the results back to those from the second experiment. Collectively, these experiments suggest that acoustic rhythms are more potent in driving behavioral modulation, with tACS serving as a supplementary modulator contingent on individual frequency preferences.

## Results

### Experiment 1

**Behavioral entrainment to FM stimuli is affected by the perceptual salience of the stimulus rhythm.** In the first experiment, we aimed to investigate whether the strength of the modulation of a rhythmic auditory stimulus (perceptual salience) affects the efficacy of tACS when both signals are presented at the same frequency, but with different phase lags (Fig 1A–1C). Participants ($N$ = 34) listened to FM sounds modulated at 2 Hz while receiving 2-Hz tACS targeting auditory regions (Fig 1B). Participants detected silent auditory targets (gaps) embedded in sounds that were presented at nine equally spaced phase locations of the FM modulation cycle. Auditory targets were presented at the individually defined detection threshold, as in previous experiments (Methods [2,31]). FM stimuli were presented at two different modulation depths (11 and 39% center to peak; Fig 1A). Each modulation depth condition was tested in a separate session, and the session order was randomized across participants. We hypothesized that tACS effects would be strongest when the modulation of the FM stimulus was weaker, under the assumption that neural entrainment to the less strongly modulated sound would be weaker and that there would be more room for the electrical signal to entrain brain signals and modulate behavior.

Seven participants were excluded from the initial sample, because they were missing one of the two sessions ($N$ = 4) or because there were no trials falling into a given combination of the FM phase and tACS lag ($N$ = 3). All results presented here correspond to data from 27 participants.

Since no measurement of brain activity was collected in this experiment, we quantified entrainment to the FM stimulus based on its behavioral readout: amplitude of the FM stimulus-induced behavioral modulation, which we call "behavioral entrainment" [2,31]. First, we estimated the effect of the stimulus modulation depth on behavioral entrainment to the sound when no active stimulation was applied (during sham). For each participant and session, gaps were binned according to the FM stimulus phase, hit rates were calculated for each phase bin, and a cosine function was fitted to the hit rates as a function of the FM phase (Fig 1D). The fitted amplitude parameter quantifies the strength of behavioral entrainment to the sound (*entAmp-FM*). Compared to single-participant surrogate distributions arrived at via a permutation-based strategy, the FM stimulus significantly modulated gap detection performance in 9/27 participants when the stimulus was presented at 11% modulation depth and 12/27 when presented with 39% modulation depth ($p$ < 0.05, uncorrected). Note that the number of participants showing significant modulation was lower than that previously reported using FM stimuli with 67% modulation, where 31/38 participants in session 1 and 34/38 in session 2 showed significant modulation [2]. As expected, behavioral entrainment to the sound (*entAmp-FM*) was stronger for the strongest (39%) than for the weakest (11%) modulated sound ($t_{(26)}$ = 2.39, $p$ = 0.024, *Cohen's d* = 0.63, Fig 1E). Thus, behavioral modulation induced by rhythmic auditory stimulation depends on the strength of sensory stimulus modulation.

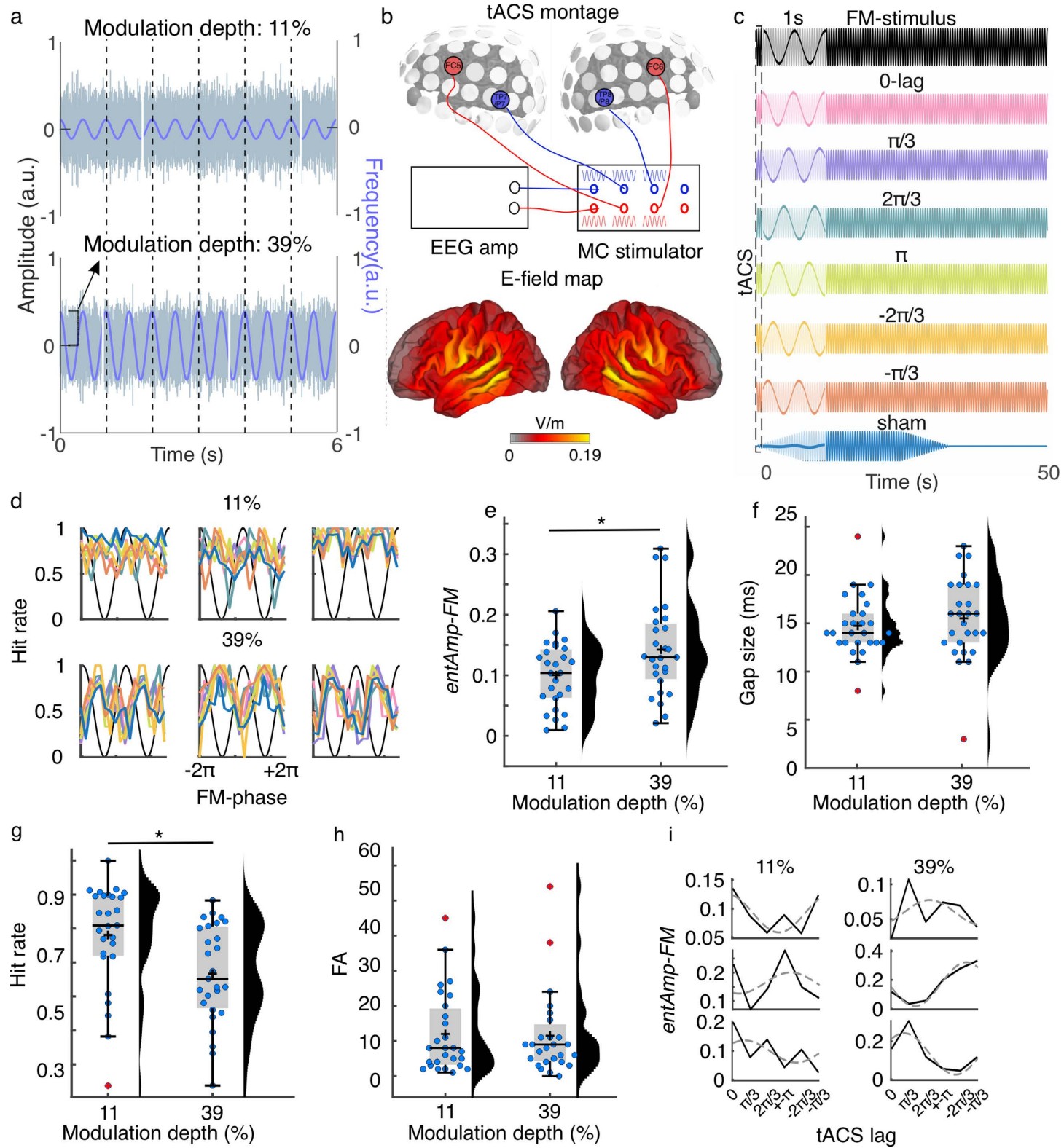

**Fig 1. Experiment 1. (a)** Participants listened to frequency-modulated (FM) auditory stimuli with varying modulation depths and were tasked with identifying silent gaps within stimuli. **(b)** Transcranial alternating current stimulation (tACS) was administered bilaterally targeting the

**auditory cortices using a multichannel system.** Concurrently, tACS-induced electrical signals were recorded via EEG for subsequent post hoc phase-lag analysis. The bottom brain maps depict the average estimated electric field distribution (E-field) of our stimulation montage across a cohort of 39 independent subjects (modified from [31], which was published under the terms of the Creative Commons Attribution License, which permits unrestricted use and redistribution provided that the original author and source are credited). **(c)** FM stimuli were initiated at random phases, producing variable phase lags in relation to the tACS signal. The trials were categorized into six phase-lag bins, constituting seven experimental tACS conditions: six specific phase lags and one sham condition. For visualization purposes, only 50 s of each stimulation condition is shown. The insets (transparent boxes at the beginning of each condition) show the same data zoomed into the first second to better observe phase lag differences. **(d)** Individual data showing hit rates as a function of the FM stimulus phase for each tACS condition when the FM stimulus was modulated at 11% (top) and 39% (bottom). Each column shows the data from a different participant. The color code is shown in (c). **(e)** Strength of behavioral entrainment to the FM stimulus (*entAmp-FM*) at each modulation depth. **(f)** Gap size threshold determined at the beginning of each session for each modulation depth. **(g)** Hit rate for each modulation depth. **(h)** Number of false alarms (FA) for each modulation depth across all stimulation conditions. **(i)** *EntAmp-FM* as a function of tACS lag for each modulation depth. Solid lines show the actual amplitude parameters obtained from the initial cosine fits of the data in (d), and dashed lines represent the second cosine fit to estimate the optimal tACS phase for modulating entrainment to the auditory stimulus. Each row represents a different participant, and the same three participants are shown for each modulation depth. **(e–h)** Each dot represents a single participant. Box plots show median (horizontal black lines), mean (black cross), 25th and 75th percentiles (box edges) and extreme datapoints not considered outliers ($\pm2.7\sigma$ and 99.3 percentiles, whiskers). Red crosses represent outliers (more than 1.5 of the interquartile range away from the bottom or top of the box). *$p = 0.05$. Numerical data for panels **d–i** can be found in [32].

No significant difference in the gap size thresholds was observed between the modulation depths ($t_{(26)} = 1.20$, $p = 0.241$, Bayes factor = 0.389, Fig 1F). During the task, hit rates were significantly higher for the 11% condition than for the 39% condition ($t_{(26)} = 3.72$, $p = 9.653e{-}04$, Cohen's $d = 0.75$, Fig 1G), suggesting that the task was easier for the 11% condition despite having similar gap sizes. By definition, the 11% sound exhibits smaller fluctuations compared to the 39% sound; that is, the peak carrier frequency is closer to the stimulus center frequency. Therefore, the higher hit rate observed in the 11% condition suggests that it is easier to detect an auditory event in a more stable environment, where the background sound undergoes less salient frequency changes. No significant difference was observed for the number of false alarms ($t_{(26)} = 0.22$, $p = 0.829$, Bayes factor = 0.208, Fig 1H, false alarms calculated across all stimulation conditions).

**tACS affects behavioral entrainment to sounds across modulation levels only at trend level.** Next, we investigated whether tACS affected behavioral entrainment to the FM stimulus, and whether this effect depended on the modulation depth of the auditory stimulus. Gaps were binned according to the FM stimulus phase and tACS lag (six lag bins), and hit rates were calculated for each FM bin and tACS lag combination (Figs 1D and S1 Fig). The strength of behavioral entrainment to the sound (*entAmp-FM*) was estimated for each tACS lag using cosine fits, as for the sham condition. Following the same procedure as in our previous study [31] we estimated the optimal tACS phase lag for each individual participant by fitting a second cosine function to the *entAmp-FM* values as a function of tACS lag (Figs 1I and S2 Fig). To facilitate analysis at the group level, which can be hindered by the high inter-individual variability in optimal tACS lags, tACS lags were realigned by making the optimal tACS phase correspond to phase zero and wrapping the remaining phases accordingly (Fig 2A–2B, [31]). To prevent bias in the analyses using the realignment procedure, *entAmp-FM* values at the new realigned peak and trough tACS phases were excluded, and their adjacent values were averaged to obtain the average *entAmp-FM* values at the optimal tACS half-cycle ($tACS_{(+)}$) and its opposite ($tACS_{(-)}$, Fig 2C–2D). Repeated measures ANOVA on these estimates (tACS condition: sham, $tACS_{(+)}$, $tACS_{(-)}$; FM modulation depth: 11 and 39%) showed a significant main effect of tACS condition ($F_{(2, 52)} = 17.83$, $p = 1.272e{-}06$, Fig 2E, left) and FM modulation depth ($F_{(1, 26)} = 7.68$, $p = 0.010$, Fig 2E, right). However, no significant interaction was observed ($F_{(2, 52)} = 0.10$), $p = 0.901$), suggesting that the effect of tACS was not significantly different across the FM modulation depths. Post-hoc multiple comparisons with Tukey's honestly significant difference criterion showed higher *entAmp-FM* values at $tACS_{(+)}$ when compared to sham (diff = 0.036, $p = 9.085e{-}4$, 95% CI = [0.014,0.058]) and to $tACS_{(-)}$ (diff = 0.045, $p = 2.633e{-}6$, 95% CI = [0.028,0.063])). No significant differences were observed between the $tACS_{(-)}$ and sham groups ($p > 0.521$). Thus, as in our previous work, we observed a significant boost in behavioral entrainment when tACS was optimally

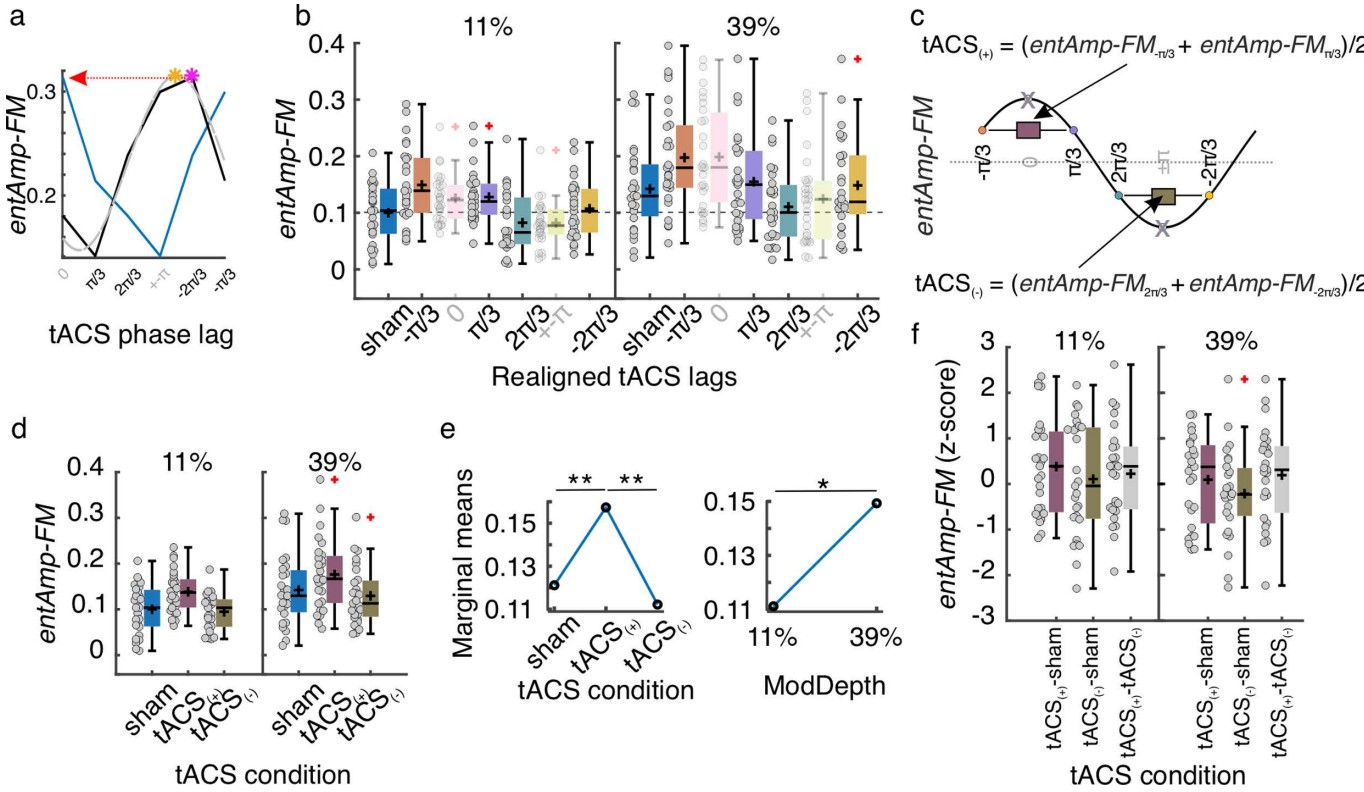

**Fig 2. Experiment 1, tACS effects. (a) Realignment procedure.** The optimal tACS lag (orange asterisk) was determined from the cosine function fitted to the *entAmp-FM* by tACS phase-lag data (dashed gray line). The phase bin closest to the optimal phase was identified (pink asterisk). This bin was set to zero phase and the remaining phases were wrapped around. The new realigned data is shown in blue. **(b)** *EntAmp-FM* as a function of tACS separated by modulation depth. The tACS lags correspond to the individually realigned lags; phase zero represents the individual optimal tACS lag estimated in (Fig 1I). *EntAmp-FM* values from the realigned peaks and troughs (semi-transparent) were excluded from further analyses. **(c)** Procedure for obtaining *entAmp-FM* at the optimal half-tACS cycle ($tACS_{(+)}$) and its opposite ($tACS_{(-)}$). **(d)** *EntAmp-FM* as a function of tACS separated by modulation depth. **(e)** Estimated marginal means, showing the main effects of the tACS condition and modulation depth (ModDepth). **(f)** *EntAmp-FM* differences between the three tACS conditions in **(c)** normalized (*z*-scores) to permuted distributions. **(b, d, f)** Each dot represents a single participant. Box plots show median (horizontal black lines), mean (black cross), 25th and 75th percentiles (box edges) and extreme datapoints not considered outliers (±2.7σ and 99.3 percentiles, whiskers). Red crosses represent outliers (more than 1.5 of the interquartile range away from the bottom or top of the box). *$p$ = 0.05, **$p$ < 0.001. Numerical data for panels **b**, **d**, and **f** can be found in [32].

aligned with the stimulus rhythm (relative to sham). In contrast to our previous work, here we did not see a shrinkage in our behavioral entrainment metric when tACS was misaligned with the stimulus rhythm, potentially because entrainment was – by design – weaker to begin with in the current study. As already observed when analyzing only the sham condition, *entAmp-FM* was higher for 39% than for 11% modulation depth (diff = 0.038, $p$ = 0.010, 95% CI = [0.010,0.066]).

To further test whether the increase in *entAmp-FM* during $tACS_{(+)}$ relative to sham and $tACS_{(-)}$ was significantly greater than chance, we created surrogate datasets by shuffling the corresponding tACS lag label for each trial while keeping all other information fixed. Following the same procedure as for the empirical data, 1,000 surrogate values of *entAmp-FM* at $tACS_{(+)}$ and $tACS_{(-)}$ were obtained for each participant and FM modulation depth. The magnitude of the tACS effect was assessed by computing the difference between *entAmp-FM* values during $tACS_{(+)}$ and $tACS_{(-)}$ conditions compared to sham (subtraction: $tACS_{(+)}$ − sham; $tACS_{(-)}$ − sham), as well as against each other ($tACS_{(+)}$ − $tACS_{(-)}$), for both the original and surrogate datasets. The values resulting from the original data were *z*-scored using the mean and standard deviation of the surrogate distributions. While *z*-score values followed a similar pattern as in our previous study [31], one-sample

*t*-tests against zero showed no significant difference for any condition (Fig 2F, all $t_{(26)}$ < 1.85, *p* > 0.075, no multiple comparison correction). Contrary to our previous results, tACS did not significantly affect behavioral entrainment to the FM stimulus when presented at these lower modulation depths.

Finally, mixed-effects logistic regression models were used to predict single-trial detection performance from the phase of the FM stimulus, phase of tACS, stimulus modulation depth, and the two-way interactions FM stimulus phase × modulation depth and tACS phase × modulation depth (S1 Table). The winning model (delta-AIC to second-best model = 4.66, S1 Table) included only the phase of the FM stimulus, suggesting that the FM stimulus phase at gap onset was sufficient to explain trial-to-trial detection performance, and that the phase of tACS did not significantly contribute to explaining behavior. To check the significance of the winning model, surrogate distributions were created by shuffling the single-gap accuracy values (0, 1) with respect to their FM stimulus phase. The same mixed-effects logistic regression models were fitted to the surrogate data, and the receiver operating characteristic curve (ROC) and area under the curve (AUC) were estimated for the original model and surrogate models. Compared to the surrogate distribution, the original model performed significantly better (*AUC* = 0.72, *p* < .001), indicating that the FM stimulus phase predicted the gap detection performance better than chance.

Overall, Experiment 1 showed that behavioral entrainment to auditory stimuli is affected by the perceptual salience of the auditory rhythm (i.e., its modulation depth), but there was no interaction between the stimulus modulation depth and tACS. On a trial-by-trial basis, the phase of the auditory stimulus was enough to explain the variability in gap detection performance. Our results confirm that auditory rhythms are superior to electrical stimulation in influencing behavior.

## Experiment 2

**Optimal tACS frequency to modulate hearing in noise shows high inter-individual variability.** In Experiment 1, we showed that FM stimuli significantly entrained behavior (gap detection performance), and the strength of this behavioral entrainment was affected by the stimulus modulation depth. Contrary to our hypothesis, we did not find convincing effects of tACS, suggesting that rhythmic auditory stimulation wins over tACS during entrainment, even when the stimulus rhythm is relatively weak. Next, we asked whether tACS could entrain behavior in the absence of competing rhythmic auditory stimulation. In Experiment 2, participants performed a gap detection task similar to Experiment 1, but gaps were embedded in narrow-band noise without any rhythmic modulation (Fig 3A). Because no rhythmic information was conveyed by the sound, making it difficult to predict the relevant frequency at which tACS should be applied, we chose four different frequencies within the delta range: 0.8, 2, 3.2, and 4.4 Hz, in addition to a sham condition (Fig 3A).

First, we evaluated whether tACS affects the overall performance of the task. Repeated measures ANOVAs showed no significant effect of tACS frequency, either for hit rates ($F_{(4, 92)}$ = 0.37, *p* = 0.828, Fig 3B, left), reaction time ($F_{(4, 92)}$ = 0.41, *p* = 0.798, Fig 3B, right), or the number of false alarms ($F_{(4, 92)}$ = 1.3, *p* = 0.274). To quantify the behavioral entrainment to tACS, gaps were grouped into six equally spaced bins according to the tACS phase at gap onset, and hit rates were computed for each bin (Fig 3C–3D). Following the same approach as in Experiment 1, when estimating behavioral entrainment to the FM stimulus, behavioral entrainment to tACS was estimated by fitting a cosine function to the individual hit rate using the tACS phase data. The amplitude of the fit quantifies the strength of the behavioral entrainment to tACS (*entAmp-tACS*), and the fitted phase parameter is the optimal tACS phase for gap detection (*prefPhase-tACS*).

To estimate the effects of tACS phase on gap detection relative to chance, surrogate distributions were created as in Experiment 1, by shuffling the gap accuracy values (1/0) across trials while keeping the tACS phase lag fixed. The same cosine fits were done as for the original data and the real *entAmp-tACS* values were *z*-scored using the individual surrogate distributions. One-sample *t*-tests were performed to evaluate the significance of the *z*-scored *entAmp-tACS* relative to chance. No significant effect was observed at the group level for any of the frequencies (all $t_{(23)}$ < 0.5, all *p* > 0.62, uncorrected).

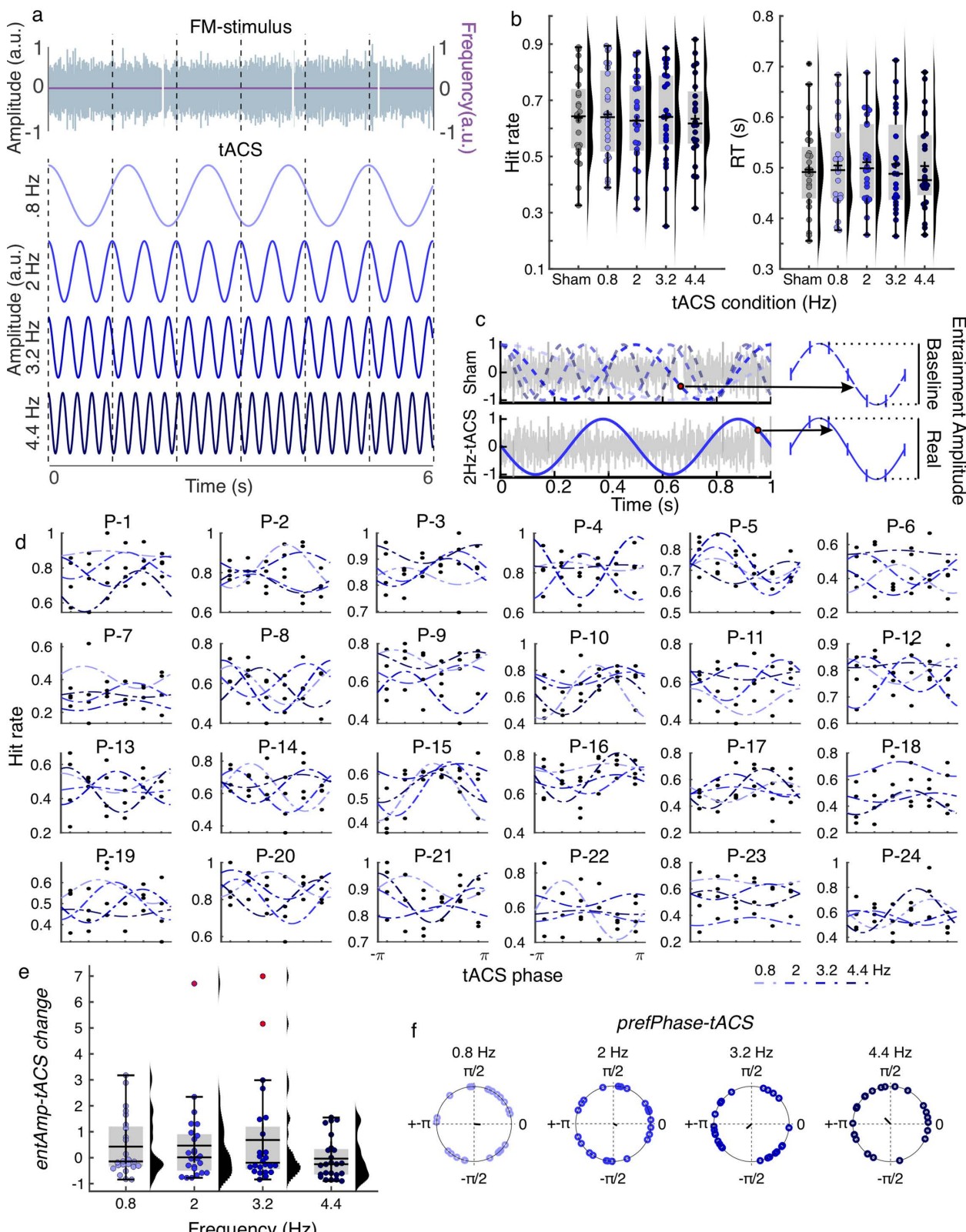

**Fig 3. Experiment 2. (a) Participants listened to complex noise stimuli and were tasked with identifying silent gaps within them.** Concurrently, tACS was applied at one of four frequencies within the delta range. **(b)** Hit rates and mean reaction times as functions of tACS condition. **(c)** For each

tACS frequency, gaps were grouped based on tACS phase. Cosine functions were fitted to the data to estimate the amplitude of behavioral entrainment to tACS. Gap detection performance at sham was used to establish a baseline level of rhythmic fluctuations. We assumed an oscillator that resets at each stimulus onset during sham. Gaps were grouped according to these hypothetical oscillators, and cosine functions were fitted to the resulting data. This was done for the four frequencies used for tACS stimulation. The amplitude parameter from these fits served as the baseline for analyzing tACS effects. In the figure, this is illustrated for the 2 Hz tACS condition. **(d)** Individual data showing hit rates as a function of tACS phase. The dashed lines indicate best-fit cosine functions. Each plot had a different participant. **(e)** Effect of tACS phase on gap detection performance. The plot shows the normalized values computed as (*entAmp-tACS* − baseline)/baseline, for each frequency. **(f)** Optimal tACS phase for gap detection (*prefPhase-tACS*) obtained from cosine fit. **(b, d, e, f)** Each dot represents a single participant. Box plots show median (horizontal black lines), mean (black cross), 25th and 75th percentiles (box edges) and extreme datapoints not considered outliers (±2.7σ and 99.3 percentiles, whiskers). Numerical data for panels **b**, **d**, and **e** can be found in [32].

Previous studies have shown that when detecting visual stimuli, behavioral performance fluctuates in a theta rhythm, suggesting that attention samples stimuli rhythmically [33,34]. To estimate the probability of observing rhythmic behavioral performance modulation independent of tACS, hypothetical ongoing oscillators at the same tACS frequencies (.8, 2, 3.2, and 4.4 Hz) were assumed to reset at the onset of the auditory stimulus in the sham condition (Fig 3C). Gaps were binned according to the phase of these hypothetical oscillators at gap onset, and cosine functions were fitted to the hit rates by oscillator phase data separated by frequency (Fig 3C). The amplitude parameter of the fits were used as the baseline amplitude of rhythmic fluctuation at each frequency in the absence of tACS. A repeated measures ANOVA was performed on the amplitude values obtained for each frequency from the tACS real stimulation and this baseline condition using a 4 × 2 repeated measures design (4 frequencies: 0.8, 2, 3.2, 4.4 Hz; 2 conditions: baseline versus tACS). No significant effects were observed for either factor frequency ($F_{(3, 69)}$ = 0.12, $p$ = 0.950), condition ($F_{(1, 23)}$ = 0.43, $p$ = 0.521), or interaction term ($F_{(3, 69)}$ = 0.538, $p$ = 0.658).

Thus, at the group level, we did not observe stronger behavioral modulation during tACS compared to baseline. To better appreciate the effect of tACS at the individual level, *entAmp-tACS* values were normalized to the baseline as (*entAmp-tACS*-baseline)/baseline (Fig 3E). Inspection of the data shows high inter-individual variability with some participants showing positive and other a negative tACS effect. Closer inspection of the data showed that 17/24 participants showed a positive tACS effect that was >1, for at least one of the tACS frequencies, meaning that *entAmp-tACS* was at least twice the baseline amplitude of rhythmic fluctuations at that frequency. The latter suggests that gap performance was modulated by tACS for part of the sample, but the relevant frequency was different across participants (see next section). On the opposite site, the negative tACS effects could potentially be due to a mismatch with the resonance frequency or ceiling effects—i.e., rhythmic behavioral fluctuations were already high at baseline, and tACS disrupted them.

In agreement with the literature, the optimal tACS phase for gap detection was randomly distributed across participants for all tACS frequencies (Rayleigh test, all $z$ < 1.177, all $p$ > 0.311; Fig 3F).

**An oscillator model reproduces individual variability of tACS effects by the mismatch between tACS frequency and individual resonance frequency.** Thus far, we have demonstrated that the optimal tACS frequency for modulating gap detection appears to be specific to the individual. This aligns with the Arnold tongue principle, which states that entrainment to an external oscillator is strongest when it matches an individual's endogenous resonance frequency [35]. We interpreted our findings as suggesting that auditory cortical oscillators in different participants were characterized by different resonance frequencies, potentially explaining the individual variability in responsiveness to tACS stimulation.

To test this hypothesis, we used the oscillator model proposed in [30] which predicts changes in entrainment to tACS as a function of the mismatch between the intrinsic oscillator resonance frequency and the stimulation frequency, as well as the amplitude of tACS (Fig 4A).

The model simulates an ongoing oscillator at a specific resonance frequency (from 0.1 to 8 Hz in steps of 0.1 Hz) and models how this oscillator is influenced by tACS at each frequency of interest (0.8, 2, 3.2, and 4.4 Hz), depending on their frequency mismatch and the relative strength of tACS to the ongoing oscillation (Fig 4A–4B). TACS amplitude took on values from 0% to 100% of the baseline oscillation's amplitude, in steps of 10%, which corresponds to physiologically

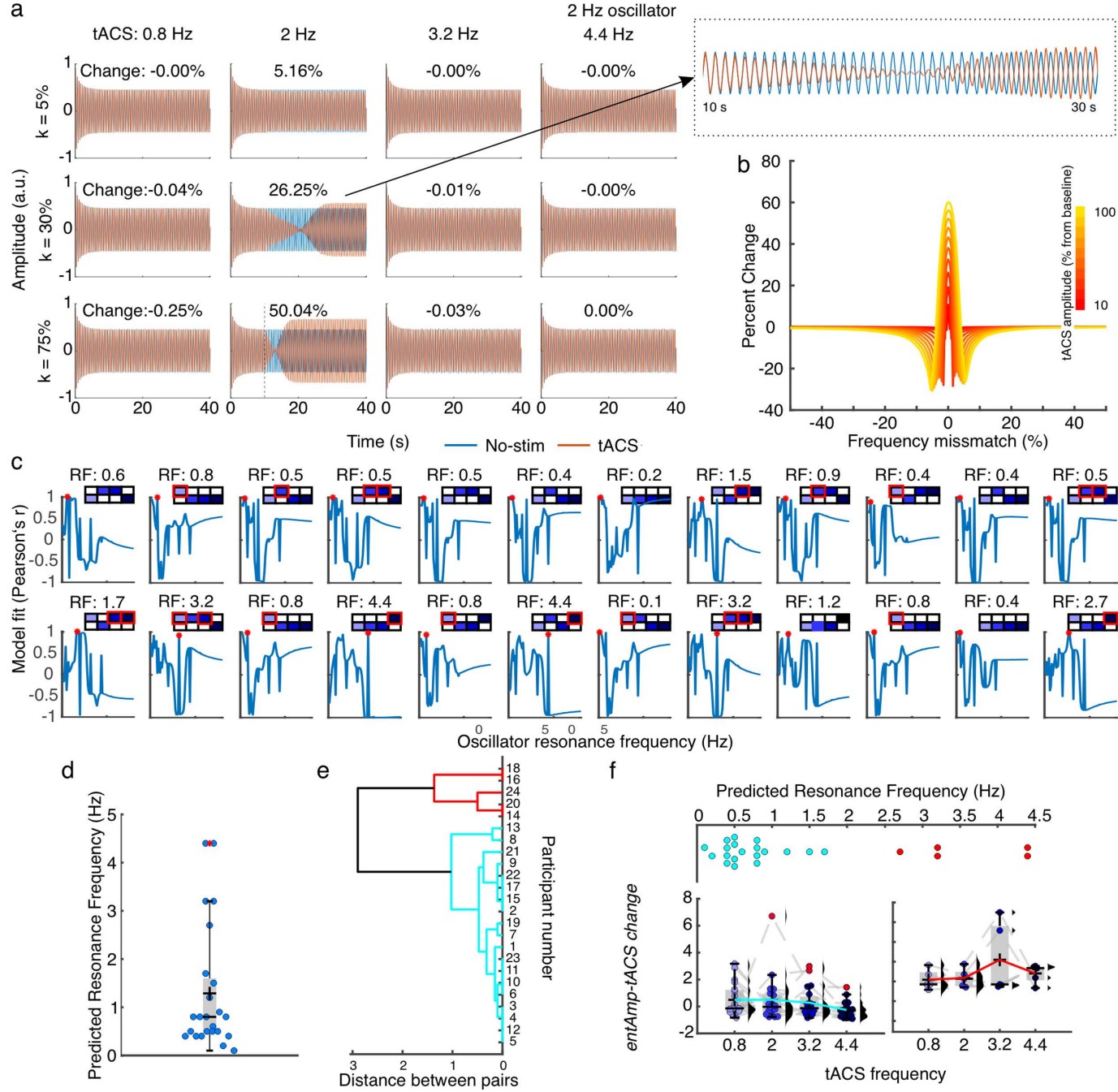

**Fig 4. Experiment 2: Oscillator model. (a) Model results for a 2 Hz ongoing oscillator at each tACS frequency, shown for three different *k*
values — tACS strength relative to the ongoing oscillator amplitude.** For illustration purposes, the tACS signal in the plots is phase-shifted by
180 degrees relative to the ongoing oscillation to better visualize entrainment. The percentage at the top of each plot represents the change in entrain-
ment due to tACS − (entrainment amplitude during tACS − baseline)/baseline*100. The dashed line indicates the start of the stimulation at time = 10 s.
**(b)** Output of the oscillator model showing the change in entrainment by tACS − (entrainment amplitude during tACS − baseline)/baseline*100 − as a
function of the tACS amplitude relative to the baseline, and the frequency mismatch between the tACS frequency and the resonance frequency of the
internal oscillator. **(c)** Individual data showing the correlation between the model output – change in entrainment relative to baseline – and the empirically
observed amplitude of behavioral entrainment to tACS − (*entAmp-tACS*-baseline)/baseline – as a function of the model's ongoing oscillator reso-
nance frequency. The boxes at the top of each plot display the effect of tACS in Experiment 2 for each individual participant. Colored squares indicate
positive (top) and negative (bottom) tACS effects relative to baseline. Red squares highlight participants with a tACS effect greater than 1, meaning
*entAmp-tACS* is at least twice the baseline amplitude. RF: resonance frequency, red asterisks. **(d)** Predicted individual resonance frequencies. Each dot

 

represents a single participant. **(e)** Dendrogram showing the clustering results. Each color represents a different cluster. **(f)** *EntAmp-tACS* values as a function of tACS frequency grouping participants based on their cluster identities in **(e)**. Spread plots at the top show the predicted individual resonance frequencies for each participant in the cluster corresponding to the plots at the bottom. Each dot represents a single participant. Box plots show median (horizontal black lines), mean (black cross), 25th and 75th percentiles (box edges) and extreme datapoints not considered outliers (±2.7σ and 99.3 percentiles, whiskers). Numerical data for panels **d** and **f** can be found in [32].

relevant conditions [30]. The amplitude of entrainment to tACS is then quantified as (tACS amplitude − baseline amplitude)/baseline amplitude. This change in oscillation amplitude relative to baseline is subsequently correlated with the change in behavioral modulation due to tACS observed in experiment 2, calculated as (*entAmptACS* − baseline)/baseline. The resonance frequency is defined as the ongoing oscillator frequency at which the model's tACS-driven entrainment change shows the highest correlation with tACS-driven behavioral modulation in the experimental data. (Fig 4C). Note that we included all participants here regardless of whether they responded positively or negatively to tACS to determine whether response patterns could be predicted by the mismatch between individual resonance frequency and tACS. This model reproduces individual patterns from the empirical data. Clustering analysis of model-based individual preferred frequencies showed two main clusters centered at lower (mean resonance frequency 1.8) and higher frequencies (mean resonance frequency 3.8) within the delta range (Fig 4C–4F). For visualization purposes, we plotted *entAmp-tACS* using frequency data separating participants by the resonance frequency cluster (Fig 4F). Each cluster was characterized by a different profile showing the strongest *entAmp-tACS* values for the tACS frequencies closest to the resonance frequencies. This analysis corroborates the hypothesis that when no rhythmic information is carried by the sensory stimulus, the strength of the tACS effects depends on the individual resonance frequency of the targeted auditory cortical oscillator.

## Experiment 3

**Optimal FM frequency for neural and behavioral entrainment varies across individuals.** Thus far, we have investigated behavioral entrainment to tACS when the electrical signal competes with a rhythmic auditory stimulus (Experiment 1) and when the auditory stimulation did not carry any rhythmic information (Experiment 2). To deepen our understanding of the competition between entrainment to auditory and electrical signals, in Experiment 3, we investigated the other side of the coin by characterizing neural and behavioral entrainment to rhythmic auditory stimulation in the absence of tACS. Our ultimate aim was to determine whether the individual resonance frequency would be similar regardless of the modality of the entraining signal: auditory or electrical. Participants, re-recruited from Experiment 2, listened to FM stimuli modulated at four different frequencies, matching the tACS frequencies from Experiment 2:0.8, 2, 3.2, and 4.4 Hz. As before, participants detected target gaps presented in different phase bins of the FM cycle (Fig 5A). Additionally, EEG data were collected while participants performed the task.

Neural entrainment to FM stimuli was investigated using frequency analysis of the EEG data. As in our previous study [2], peaks in the amplitude spectra were observed at most FM frequencies, with the exception of 0.8 Hz (Fig 5B). As another indicator of entrainment, we computed the resultant vector length over trials, which is an index of intertrial phase coherence. Here, peaks characterize the activity that is phase-aligned across trials and with the auditory stimulus. Vector length peaks were also observed at the FM frequency and first harmonics, with the exception of the 0.8 Hz condition (Fig 5C).

To quantify the effect of FM frequency on neural entrainment, amplitude and vector length values were extracted for each participant and averaged across a cluster of fronto-central electrodes (F3, Fz, F4, FC1, FCz, FC2, C3, Cz, and C4), in line with our previous study characterizing neural entrainment in the auditory system [2]. This cluster of electrodes represents the typical auditory topography in EEG data [2]. To control for different noise levels in estimating the amplitude and vector length at different FM frequencies (1/*f* noise, methodological limitation in estimating the peaks, other measurement noise), the amplitude and vector length values were normalized by dividing the values observed at each FM frequency

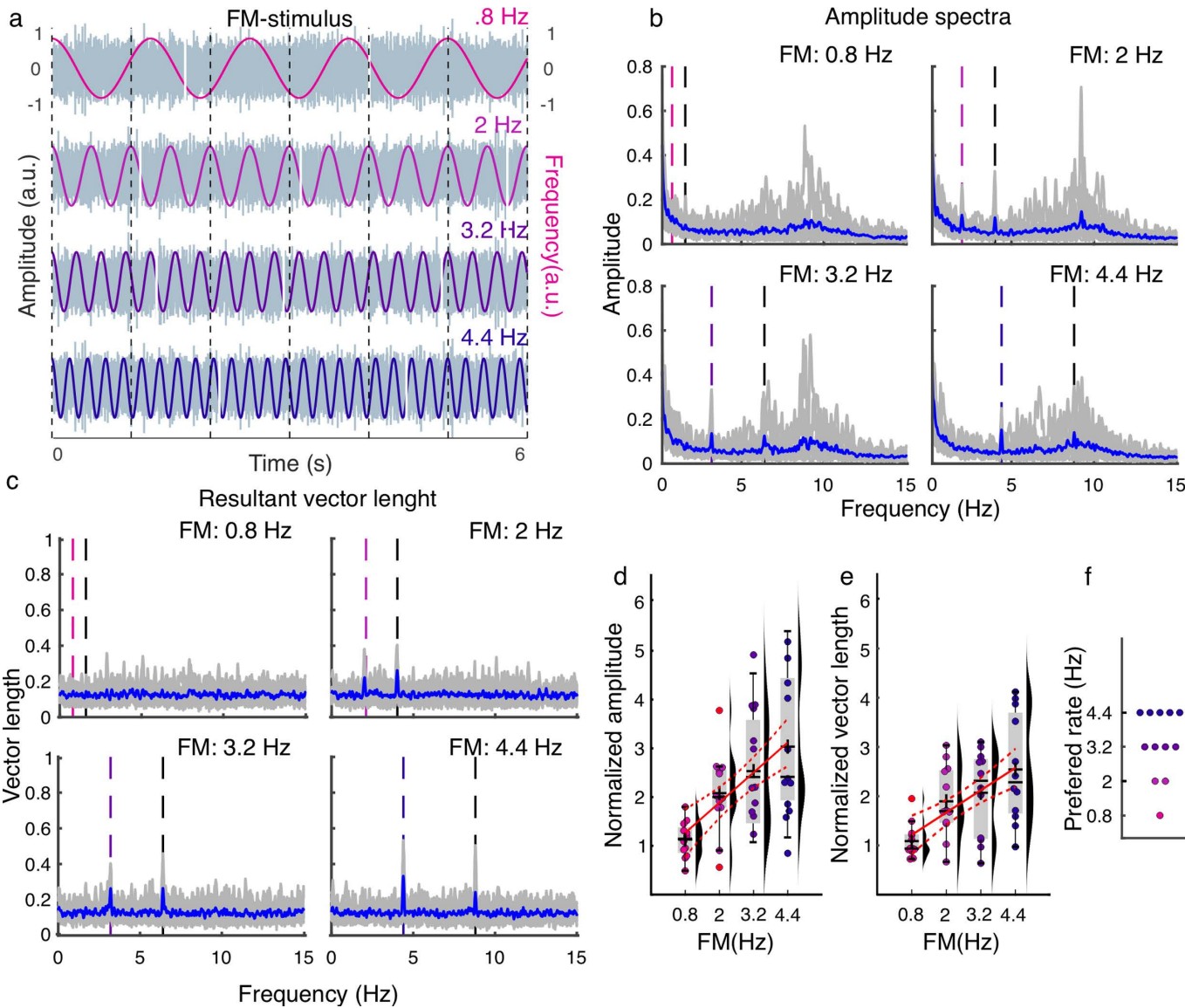

**Fig 5. Experiment 3, EEG. (a) Cartoon depicting the stimulus design.** Participants listened to sounds modulated at different frequencies and detected gap periods presented in different phase bins of the modulation cycle. **(b)** EEG Amplitude spectra separated by FM frequency. The dashed vertical lines mark the FM frequency (in color) and first harmonic (black). Average group data are shown in blue, whereas individual data are shown in gray. **(c)** Resultant vector length spectra separated by the FM rate. The colors follow the same convention as in **(b)**. **(d)** Normalized amplitude values quantifying neural entrainment to the FM stimulus separated by frequency. The amplitude values in **(b)** were normalized by dividing the amplitude of the EEG activity observed at each FM frequency during the presentation of the FM stimulus at a specific frequency by the mean amplitude recorded at that frequency when the FM stimulus was presented at all other frequencies. **(e)** Normalized resultant vector length as a function of FM frequency. Normalization was performed as described in **(d)**. **(f)** Optimal FM frequency (rate) for neural entrainment (FM frequency with the highest normalized resultant vector length). **(d–f)** Each dot represents a single participant. **(d–e)** Box plots show the median (horizontal black lines), mean (black cross), 25th and 75th percentiles (box edges), and extreme data points not considered outliers (±2.7σ and 99.3 percentiles, whiskers). Red crosses represent outliers (more than 1.5 of the interquartile range away from the bottom or top of the box). Solid red lines display the best fit from the linear model, and dashed red lines show the 95% confidence intervals. Numerical data for panels d and e can be found in [32].

during the presentation of the FM stimulus at a specific frequency by the mean amplitude recorded at the same frequency when the FM stimulus was presented at all other frequencies. To test for the effect of FM frequency, two linear models were fitted to the normalized amplitude and vector length values using FM frequency as a linear regressor. This analysis showed that both normalized amplitude and vector length values linearly increased as a function of FM frequency (Fig 5D–5E, normalized amplitude: $R^2$ = 0.331, *Adjusted* $R^2$ = 0.317, *F-statistic versus constant model* = 22.8, *p* = 1.87*e*−05; normalized vector length: $R^2$ = 0.295, *Adjusted* $R^2$ = 0.28, *F-statistic versus constant model* = 19.2, *p* = 6.65e−05).

Each participant's preferred FM rate for neural entrainment was identified as the FM frequency that induced the highest normalized vector length value. The participants' preferred frequencies were spread across all tested FM frequencies (Fig 5F), with a small number of individuals in each category (1/12 preferred 0.8 Hz, 2/12 preferred 2 Hz, 4/12 preferred 3.2 Hz, and 5/12 preferred 4.4 Hz). This suggests that as for tACS, the optimal frequency for neural entrainment to rhythmic sensory stimulation varies across participants.

Subsequently, we explored the effect of the FM stimulus frequency on behavioral entrainment (Fig 6). Previously, we demonstrated that hit rates decline as a function of the frequency modulation (FM) rate when a fixed gap duration is employed [2]. To minimize the possibility that these differences in detection rates could affect the estimation of behavioral entrainment, we adjusted the gap durations to threshold levels (approximately 50% accuracy) individually for each participant and FM rate. While the gap duration threshold appeared to increase in relation to FM rate, this effect did not reach statistical significance (Fig 6B, linear regression model, $R^2$ = 0.052, *Adjusted* $R^2$ = 0.031, *F-statistic versus constant model* = 2.5, *p* = 0.12).

Despite our efforts to equalize task difficulty across FM rates, hit rates significantly decreased with FM rate (Fig 6C, *linear regression model:* $R^2$ = 0.231, *Adjusted* $R^2$ = 0.214, *F-statistic versus constant model* = 13.8, *p* = 5.480*e*−04). The false alarms rate was low (mean = 6.92 across conditions) and was not affected by the FM rate (*linear regression model:* $R^2$ = 0.0145, *Adjusted* $R^2$ = −0.00696, *F-statistic versus constant model* = 0.675, *p* = 0.416). To estimate behavioral entrainment to the FM stimulus (*entAmp-FM*), we fitted cosine functions to the hit rate using FM phase data, as in Experiment 1. We observed that the strength of sound-induced behavioral modulation increased with FM rate (Fig 6D, linear regression model, $R^2$ = 0.165, *Adjusted* $R^2$ = 0.147, *F-statistic versus constant model* = 9.11, *p* = 0.004). No significant difference was observed in the optimal FM phase for gap detection between FM rates (multiple parametric *Hotelling* paired sample tests for equal angular means, all $F_{(2,10)}$ < 5,08, *p* > 0.180, corrected for 6 comparisons with *Bonferroni method*, Fig 6E, [2]). To estimate each participant's preferred FM rate for behavioral entrainment, we determined the FM rate at which each individual exhibited the strongest amplitude of behavioral entrainment. The participants' preferred frequencies were 2 (*N* = 3), 3.2 (*N* = 5), and 4.4 Hz (*N* = 4) (Fig 6F), suggesting that the optimal frequency for behavioral entrainment to rhythmic sensory stimulation varies across participants.

**Optimal frequency for behavioral and neural entrainment do not overlap.** We showed that the optimal frequency for entrainment to both tACS and FM stimuli varies across participants. To close the loop, we asked if the preferred frequencies were stable within an individual, regardless of the modality of the rhythmic stimulus. S3 Fig shows the amplitude of behavioral and neural entrainment to tACS and the FM stimulus, respectively, for the 12 participants who took part in both Experiments 2 and 3. While some participants exhibit similar response profiles across tasks and measures, most show different frequency preferences (i.e., the frequency with the highest entrainment) across tasks. This suggests that the optimal entrainment frequency may depend on the stimulation modality. However, note that we tested only four frequencies and included only 12 participants in this analysis. Unfortunately, it was not possible to re-recruit more participants for testing because of the unavailability of participants and relocation of the research group. This remains speculative and requires further investigation with larger sample sizes and measuring more stimulation frequencies.

## Discussion

In this study, we investigated neural entrainment to rhythmic auditory and electrical stimulation when both signals were presented simultaneously, competing for entrainment, or independently. Participants detected near-threshold silent gaps

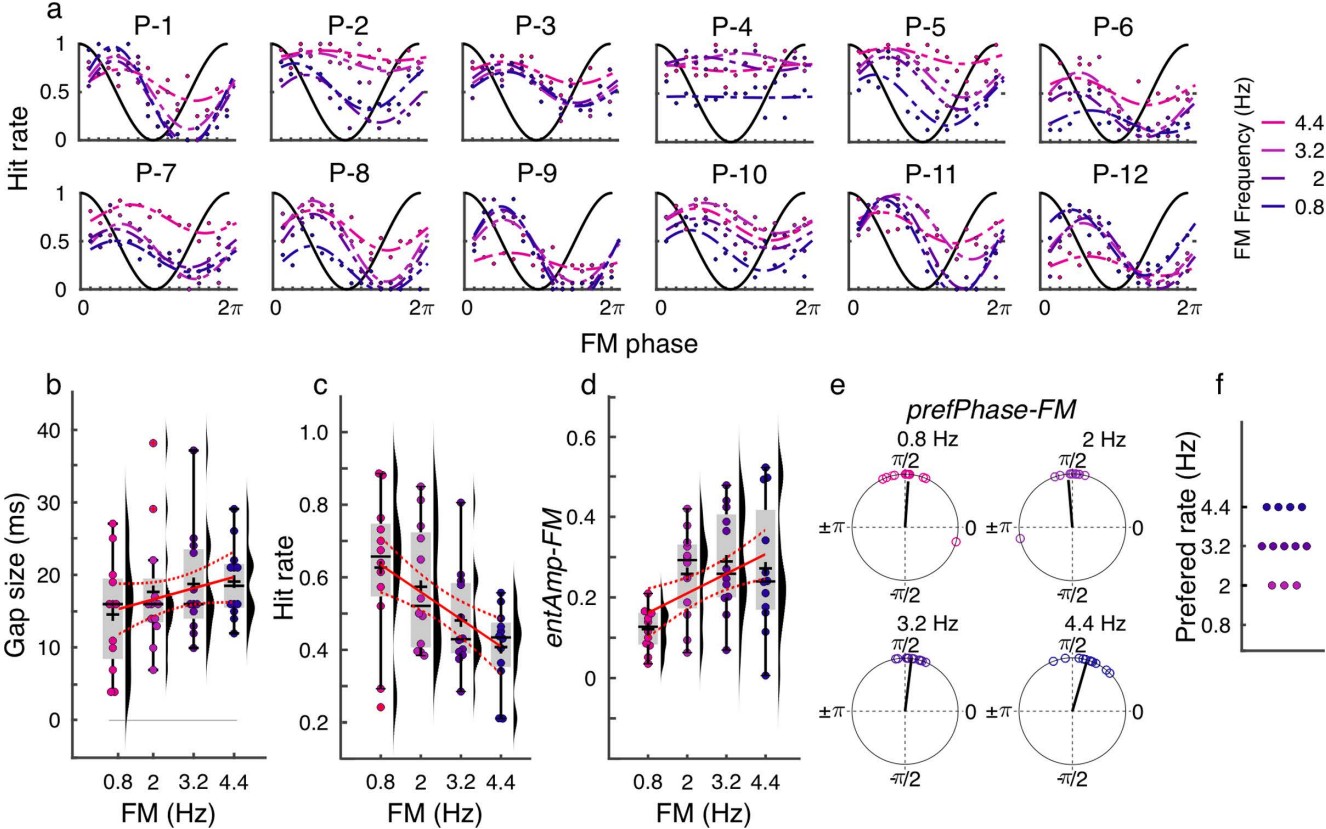

**Fig 6. Experiment 3: Behavior.** (a) Individual data showing hit rate as a function of the FM stimulus phase separated by FM frequency. The dashed lines represent the cosine fits. Each plot had a different participant. (b) Gap size threshold as a function of FM frequency. These thresholds were determined using a staircase procedure at the beginning of each session. (c) Hit rate as a function of FM frequency. (d) Strength of behavioral entrainment to the FM stimulus (*entAmp-FM*) as a function of FM frequency. (e) Optimal FM phase of gap detection performance (*prefPhase-FM*) for each FM frequency. (f) Preferred FM rate for behavioral entrainment (FM frequency inducing the highest *entAmp-FM* in (d)). (b–f) Each dot represents a single participant. (b–d) Box plots show the median (horizontal black lines), mean (black cross), 25th and 75th percentiles (box edges), and extreme data points not considered outliers (±2.7σ and 99.3 percentiles, whiskers). Red crosses represent outliers (more than 1.5 times the interquartile range away from the bottom or top of the box). Solid red lines display best fit from the linear model and dashed red lines show the 95% confidence intervals. Numerical data for panels **b–d** can be found in [32].

embedded in noise stimuli across three experiments. In Experiment 1, we assessed whether the effects of tACS on behavioral entrainment to sounds were influenced by the modulation depth of the rhythmic auditory stimulus. Experiment 2 focused on evaluating the effect of tACS frequency on modulating behavior when no rhythmic information was conveyed by the sounds, and explored the relevance of the individual resonance frequency of a hypothetical auditory cortical oscillator using a computational model. In the final experiment, conducted without tACS, we examined whether neural and behavioral entrainment to sounds were affected by the auditory stimulus rate; participants in Experiment 3 were re-recruited from Experiment 2 so that we could examine the consistency of estimated individual resonance frequencies under different modalities.

Our main findings were: (1) The salience of rhythmic auditory stimulation (modulation depth) significantly affected behavioral entrainment to sounds. (2) However, the degree to which tACS modulated behavioral entrainment to the sound stimulus did not depend on modulation depth. Regardless of modulation depth, the auditory stimulus had a greater influence on behavior than electrical stimulation. (3) When no rhythmic information was conveyed by the sound, 17 out of

24 participants exhibited significant behavioral entrainment to tACS, though the specific frequency that affected behavior most strongly varied across participants. This pattern was captured by an oscillator model with a free parameter for individual resonance frequency of the hypothetical auditory cortical oscillator. (4) In the absence of tACS, both neural and behavioral entrainment to rhythmic sounds depended on the auditory stimulus rate in an idiosyncratic way across participants. (5) Comparing results across experiments suggested that the optimal entrainment frequency may depend on the stimulation modality. Because of limitations with re-recruitment, this last result should be further investigated with bigger sample sizes. In the following sections, we will discuss each point independently, providing a deeper analysis of these findings and their implications.

### Behavioral entrainment to sounds is affected by the modulation depth of the auditory stimulus

In agreement with previous research, our results demonstrated neural and behavioral entrainment to rhythmic auditory stimulation [2]. As expected, in Experiment 1, the modulation depth of the auditory stimulus significantly affected behavioral entrainment to the sound, with stronger modulation depth resulting in stronger behavioral entrainment compared to the weaker modulation depth. Moreover, the mean strength of behavioral entrainment observed in Experiment 1 with modulation depths of 11% and 39% was smaller than that observed in Experiment 3 (compare Figs 1E and 6D) and our previous work [31], both using a modulation depth of 67%. Behavioral entrainment strength indeed monotonically increases with the modulation depth of the auditory stimulus (at least up to 67%). Previous research has shown that more strongly modulated FM signals elicit stronger neural responses [36]. This effect may occur because stronger FM signals lead to wider coverage of the frequency-tuned neuronal populations activated by the sound, due to the tonotopic organization of the auditory system. Columns of auditory cortical neurons that are broadly tuned to sound frequency respond with faster onset-offset responses to dynamically frequency- and amplitude-modulated sounds [37]. Theoretically, the FM sounds with larger modulation depths used here could more strongly activate the broadly tuned columns that efficiently follow temporal modulations. Moreover, the modulation depth in our FM stimulation corresponds to perceptual salience or intensity. More salient stimuli can capture and hold attention more effectively, potentially keeping the listener focused on the rhythmic aspects of the sound. Therefore, our results align with previous literature showing that more intense signals elicit stronger entrainment than weaker ones [35].

Of note, in our case, higher entrainment to the 39% modulation depth condition did not correspond to higher behavioral performance over all. In fact, average hit rates were higher for the 11% condition than for the 39% condition. Given the growing body of literature linking entrainment to auditory perception, at first glance one would expect that higher entrainment during the 39% modulation depth condition should be associated with better performance [3,4,6–12]. However, the relationship between entrainment and auditory perception can be task-dependent. Based on our previous EEG study [2], we know that neural activity entrains to the FM stimulus. Entrainment of slow oscillations to the FM stimulus rhythm suggests that targets presented during the excitatory phase of the entrained oscillation are better processed than those presented during the inhibitory phase. In our task, gaps are randomly presented at equally spaced bins across the modulation cycle. We characterized behavioral entrainment using the amplitude parameter of a cosine function fitted to the hit rates (performance) as a function of stimulus phase. Higher entrainment in this context reflects a larger difference between hit rates at the peak and trough of the stimulus phase—consistent with neural entrainment to the sound—and is independent of mean performance, which represents hit rates averaged across all phases.

Previous literature suggests that gap detection is better in unmodulated than in modulated noise. For example, gap detection thresholds in unmodulated broadband noise have been reported to be around 2–3 ms in early studies [38–40]. In contrast, gap detection thresholds in amplitude-modulated sounds appear to be higher, around 8–10 ms [41], and can vary further depending on the type of modulation [42]. We believe that overall task performance is higher at the 11% modulation depth because the task is easier under this condition. It is easier to detect the silent target within a stimulus that is perceptually more constant (lower modulation depth) than in the strongly modulated 39% condition. As we showed

previously [2], gap detection performance in this paradigm is influenced not only by entrainment but also by alpha power. It is possible that the continuous mode mediated by alpha activity provides a more effective mechanism than neural entrainment for this particular task.

## Rhythmic auditory signals prevail *over* electrical stimulation *to* entrain behavior

In our previous work, we found that behavioral entrainment was more strongly driven by auditory than electrical rhythms [31]. Here, we hypothesized that if we reduced the salience of the sound rhythm by decreasing modulation depth, it would give tACS more of a chance to entrain the active oscillation. To our surprise, we did not find any increase in the efficacy of tACS to modulate entrainment when we reduced the sounds' modulation depth in Experiment 1. In fact, when compared to surrogate distributions, we failed to observe a significant main effect of tACS over and above the effect that would have been expected by chance. Moreover, in agreement with our previous study [31], the trial-by-trial phase of the FM stimulus was the best predictor of gap detection performance, and the phase of tACS did not add predictive power to the model predicting trial-by-trial behavioral performance.

The lack of significant effect of tACS in this experiment contrasts with our previous study, where we showed that tACS significantly modulates entrainment to FM stimuli [31]. However, some methodological differences could help to explain this disagreement. By design, the main difference between the studies was the modulation depth of the stimulus: 11% and 39% here, compared to 67% in [31]. Our hypothesis was that weaker modulation depths should lead to less competition between the sound and the electrical signal for entrainment; therefore, tACS effects would be expected to be stronger than in our previous study. However, as evidenced by the strength of the behavioral entrainment effects we observed here, weaker modulation depths likely induce weaker neural entrainment to the sound. Our current results suggest that there might be a critical level of oscillation that needs to be driven by the stimulus in the first place for tACS to interface with it. We hypothesize the existence of a U-shape relationship between the strength of oscillatory activity (either ongoing or sensory-driven) and the ability of the electrical stimulation to affect the system. In other words, tACS would be more effective when there is some level of oscillatory activity at a frequency matching the tACS frequency (importance of the resonance frequency) but still weak enough to be moved by the electrical signal. When the state is too active, tACS might be too weak to override entrainment, when the state is not active enough, tACS would be effective only when stimulating at the system's resonance frequency. Previous studies have shown support for both extremes of this U-shape. Based on the Arnold tongue principle [35,43], rhythmic stimulation is more likely to entrain a system when it matches its resonance frequency. However, we know that tACS competes with ongoing oscillations for entrainment and that tACS effects are strongest when the power at the relevant frequency is not too high [30,44]. It could be that in Experiment 1, we did not reach the minimum level of behavioral entrainment at 2 Hz to make 2 Hz tACS signal effective. In this scenario, we expect it would be more effective to apply tACS at the individual resonance frequency, rather than using the same 2 Hz frequency for every participant as we did (see further discussion about the Arnold tongue principle in following subsections). This reasoning would reconcile the results from this and our previous study [31]. We did not a priori know the resonance frequency of any individual's auditory cortical oscillator. Future studies could test the impact of stimulating at the individual resonance frequency on tACS effects during concurrent auditory stimulation with various modulation depths.

Another methodological difference to our previous study was that in the current study, the final sample size was smaller than in the previous one, and only one session per modulation depth was recorded, whereas the previous study tested two sessions, leading to double the data per participant. Given the variability inherent in tACS effects, more data might be needed to demonstrate a significant effect. To clarify this point, we conducted the statistical analysis using the pre-processed data from our previous study [31], randomly selecting different sample sizes and using either one or both sessions. S4 Fig confirms that the probability of observing a significant effect increases with sample size and is higher when data from multiple sessions is included. This analysis suggests that a minimum of 32 participants and two sessions is required to detect a significant difference in behavioral entrainment to the FM stimulus between positive and negative half-cycles of tACS with 0.95 probability. However, this number may vary depending on the specific task.

Last, the lack of tACS effects could be due to the use of a one-size-fits-all electrode montage in this experiment, whereas half of the participants in our previous study had individualized tACS montages. Although we did not observe any significant difference in the magnitude of tACS effects between montages in our previous study, the results suggested lower inter-individual variability for the individualized montage group [31]. Therefore, using the same montage for every participant in the current experiment might have reduced the likelihood of observing a significant effect due to inter-individual variability. Further investigation into the superiority of individualized tACS montages to modulate behavior, possibly in within-subject designs, would also be a step forward in the field.

Contrary to our hypothesis, we did not observe any significant interaction between the modulation depth of the auditory stimulus and tACS. In our previous study, we showed that tACS effects were only modulatory and that the auditory stimulus exerted the strongest effect on behavior [31]. Based on previous literature suggesting that the strength of ongoing oscillations negatively relates to the effects of tACS [30,44], indicating competition between endogenous neural oscillations and the electrical signal, we hypothesized that entrainment to the clear rhythm of the FM stimulus might have been too strong for tACS to overwrite. Therefore, tACS effects could only be modulatory. Phasic effects of tACS on auditory perception have primarily been observed during auditory stimulation with speech stimuli whose amplitude envelope had been removed [26,27] and in the context of other weakly modulated background sounds [22]. We predicted that tACS would be more likely to shift the phase of neural entrainment when entrainment is weaker to start with, either due to weaker stimulus modulation or in listeners lacking young, healthy auditory systems [45]. Here, we tested the hypothesis that weaker stimulus modulation would result in stronger tACS effects, anticipating that tACS effects should be stronger for the 11% modulated sound than for the 39% modulated sound, as weaker auditory stimuli should represent lower competition to the electrical signal for entraining the brain and behavior. However, we did not find evidence for this.

Our second hypothesis was that rhythmic sensory stimulation would be superior to electrical signals for entraining the brain and modulating behavior [31]. When both the auditory stimulus and the electrical signal are rhythmic and compete for entrainment, the auditory stimulus might always prevail as long as the rhythmic information can be perceived, regardless of the modulation depth of the stimulus. Our current results support this interpretation.

There are several reasons why FM (frequency-modulated) stimuli might be more effective than tACS in entraining behavior. First, our study utilized an auditory task involving near-threshold gap detection. Consequently, both the entraining stimulus and the target to be detected are in the same sensory modality. FM sounds can efficiently synchronize neuronal activity in auditory regions [4,36]. In contrast, tACS applies weak electrical currents through the scalp, which may be less specific to the auditory cortex depending on the stimulation parameters [18]. This broader targeting can reduce the precision of auditory entrainment compared to the focused FM stimulation. Given that both the entraining stimulus and the target are presented in the same modality, it is likely easier for the sound to influence behavior. A follow-up question to consider is whether rhythmic sensory stimulation is more effective than tACS regardless of the sensory modality, or if this effect is specific to within-modality conditions. Previous research has shown that rhythmic sensory stimulation can entrain higher-order areas [46] and influence target detection in a different modality from the entraining stimulus [47]. However, to our knowledge, there are no tACS studies using these kinds of rhythmic sensory stimulation.

A key challenge in entraining behavior with two different signals is the difficulty of comparing their intensities. Assuming that auditory and electrical signals compete to reset the phase of an intrinsic oscillator, the extent of phase resetting depends on the intensity of the perturbation. While the intensities of both entraining signals were kept constant throughout the experiment, determining whether these intensities are equivalent is challenging. The intensity of the auditory stimulus was individually defined to be 55 dB above the hearing threshold, whereas the intensity of tACS was fixed at 1 mA peak-to-peak for all participants. A major limitation is the lack of a direct measure of tACS efficacy on the excitability of the auditory system. Unlike the motor or visual systems, where proxies such as motor-evoked potentials or phosphene thresholds are used to gauge cortical excitability, no comparable metrics exist for the auditory system. One potential solution could involve calibrating tACS intensity relative to an individual's detection threshold using skin sensation as an indirect proxy.

Although skin sensation from tACS does not directly correspond to auditory stimulus thresholds, it could help account for individual variability in perception. Note that most participants in our study reported to feel the stimulation to some extent, which suggests that the stimulation intensity of 1 mA was above the skin sensation threshold. Another promising approach, recently proposed for TMS [48], involves defining stimulation intensity based on the motor threshold during motor cortex stimulation and estimating the corresponding intensity in the target area using computational modeling. Future research could explore and validate the applicability of this method for tACS. Our choice of 1 mA for tACS stimulation was guided by safety considerations, given the electrode size, as well as tolerability for the average participant. While many tACS studies have utilized higher intensities (e.g., 1.5–2 mA), prior research has demonstrated that 1 mA tACS is also effective for modulating auditory perception [31,49] and alter brain activity and functional connectivity [50,51]. Our findings suggest that auditory rhythms presented well above the hearing threshold are more effective than 1 mA tACS at entraining behavior. Future research could further investigate this relationship and explore alternative methods to standardize and compare the intensities of auditory and electrical signals. On this line, it would be interesting to examine whether the efficacy of tACS can be enhanced by increasing the stimulation current strength or by weakening the auditory signal, not in terms of modulation depth as in our study but in terms of sound level. The former approach is limited by the safety thresholds of tACS stimulation, while the latter is more feasible to test. Although these aspects were beyond the scope of our study, they should be explored in future research.

### tACS effects during non-rhythmic auditory stimulation depend on hitting the individual resonance frequency

In our second experiment, where no rhythmic information was conveyed by the sound, gap detection behavior was significantly modulated in 17 out of 24 participants. However, the modulation did not consistently occur at the same tACS frequency across all participants, suggesting individual variability in preferred frequency for entrainment to electrical signals. This finding was further supported by an oscillator model with a free parameter for individual resonance frequency, which produced similar individual modulation profiles as we observed in our behavioral data. The model indicates that the strength of behavioral entrainment to tACS is dependent on the alignment between the stimulation frequency and the individual's endogenous resonance frequency. As already mentioned, the concept of 'Arnold tongue' is relevant here: entrainment to an external driving rhythm is most effective when it matches an individual oscillator's endogenous resonance frequency [31]. This pattern has been demonstrated for neural entrainment in the context of rhythmic visual stimulation [43]. Our findings align with this principle and suggest that the resonance frequency of intrinsic oscillators in the auditory system varies among individuals. Previous studies investigating tACS effects at the neural level have shown the Arnold tongue effect [52], indicating strongest entrainment when external stimulation aligns with intrinsic frequencies. However, to our knowledge, our study is among the first to demonstrate this effect in the context of auditory behavior. This extends the applicability of the Arnold tongue principle from neural entrainment to behavioral modulation and highlights the importance of considering individual differences in resonance frequencies within the delta range when applying tACS during auditory tasks, something that is more common to do in the context of alpha and gamma tACS during visual tasks (see, e.g., [44]).

Future research should explore methods to personalize tACS frequencies to match individual resonance frequencies more accurately. Additionally, investigating the underlying neural properties that contribute to the variability in resonance frequencies could provide further insights. Understanding these properties may lead to improved strategies for using tACS in therapeutic settings, potentially enhancing its efficacy for auditory and other sensory modulations.

### Neural and behavioral entrainment to rhythmic sounds is affected by the stimulus rate

The final experiment focused on neural and behavioral entrainment to rhythmic sounds without the influence of tACS. The results indicated that both neural and behavioral responses were modulated by the rate of the auditory stimulus. However, the optimal stimulus rate varied across participants, suggesting individual differences in auditory rhythm processing. These findings highlight the importance of considering individual variability in neural processing and responses to rhythmic

stimuli. The effect of stimulus rate on neural entrainment has been shown previously in the literature, for example in the context of speech stimuli and auditory click trains [53]. Interestingly, in their study, neuronal synchronization to speech was enhanced at 4.5 Hz while neural entrainment to clicks showed a different pattern linearly decreasing for lower frequencies and peaking at 8.5 Hz. In this study, we used FM sounds and showed that neural and behavioral entrainment linearly increase as a function of the stimulus rate. Note however that we did not test frequencies higher than 4.4 Hz. Therefore, it is not possible to estimate whether 4.4 Hz would be the peak frequency for entrainment to FM sounds, similarly to speech, or if entrainment would continue increasing had we tested higher frequencies. The fact that both neural and behavioral entrainment to 0.8 were the weakest in our study, suggests that the lower cut for entrainment to FM-stimuli could be somewhere between 0.8 and 2 Hz. It remains to be tested what the peak frequency and upper cutoff frequency would be. Nevertheless, answering this question was outside the scope of our study.

### Preferred rate for behavioral and neural entrainment might be independent

Given the individual differences in effectiveness of different tACS frequencies to modulate different individuals' behavior, we wanted to test whether there would be a correspondence to frequencies of auditory rhythms that best drove either behavioral or neural entrainment. Since we re-recruited participants from Experiment 2, we were working with a relatively small sample size, so this section remains speculative. A notable observation from this study is the lack of clear alignment between individual preferred frequencies for neural entrainment to sounds, its behavioral readout, and behavioral entrainment to tACS. When examining individual response profiles across tasks, we observed that while a few participants exhibited similar response patterns across tasks and measures, most displayed different frequency preferences (i.e., the frequency with the highest entrainment) across tasks. This discrepancy may suggest that the mechanisms underlying neural and behavioral entrainment to electrical and auditory stimulation may differ, and in turn that behavioral modulations may not exclusively be a result of neural entrainment but a combination of different mechanisms. The divergence in preferred frequencies emphasizes the need for personalized approaches in the application of tACS for therapeutic or cognitive enhancement purposes. While a possible (and logical) strategy would be to estimate optimal frequencies from EEG, the discrepancy we observed between neural entrainment to sounds and behavioral entrainment to tACS suggests that this might be unfruitful, at least in the context of auditory stimulation with FM sounds. Estimating preferred frequencies from behavioral measures instead of from EEG might be more appropriate if the effects of tACS would also be measured at the behavioral level. Overall, unpacking the relationship between neural and behavioral entrainment to sound and electrical rhythms will require future work with larger sample sizes.

### Summary and implications

In conclusion, this study provides a comprehensive analysis of entrainment to rhythmic auditory and electrical stimulation, revealing important nuance and individual variability. In particular, the variability in individual resonance frequencies highlights the necessity of personalized interventions when using tACS for clinical or experimental purposes. On the other hand, our results indicate that those resonance frequencies may not be easily measured using EEG (though behavioral entrainment measures may be a more promising route). We suggest that future research should explore the underlying neural mechanisms contributing to the observed differences in, and structural/functional correlates of, entrainment preferences and how these can potentially inform optimization of tACS protocols based on individual resonance characteristics. Results show that rhythmic auditory stimulation dominates over tACS for modulating auditory perception, which opens new avenues for non-invasively modulating the brain with sensory stimulation. This would be an easier-to-use and less expensive approach than conventional brain stimulation protocols. Overall, the findings contribute to the broader understanding of how the brain processes competing stimuli and offer valuable insights for the development of tailored neuromodulation therapies.

## Materials and methods

### Ethics statement

All procedures were approved by the Ethics Council of the Max Planck Society (application nos. 2019_04 and 2019_27 for the EEG and tACS experiments, respectively). Participants provided written informed consent and received financial compensation for their participation.

### Participants

Fifty healthy participants took part in the study (31 females, 16 males and 3 divers, mean age = 27.02; SD = 4.66, range: 18–38). Thirty-four participants took part in Experiment 1 (34 in one tACS session and 30 in two tACS sessions separated by 3–301 days (median: 7 days). Twenty-four participants (8 overlapped with Experiment 1) took part in Experiment 2 (one tACS session), 12 of which also joined Experiment 3 (one EEG session). All participants self-reported normal-hearing and normal or corrected-to-normal vision. At the time of the experiment no participant was taking medication for any neurological or psychiatric disorder.

### Experiment 1

**Auditory stimuli.** The auditory stimuli were created using MATLAB 2017a (MathWorks) with a sampling rate of 44.1 kHz. These stimuli consisted of 20-second-long complex tones, modulated in the frequency domain at 2 Hz, with a peak-to-center modulation depth of 11% and 39% (Fig 1A). Each stimulus had a randomly selected center frequency between 1,000 and 1,400 Hz. The complex tone comprised 30 components sampled from a uniform distribution with a 500-Hz range around the center frequency. The amplitude of each component decreased linearly as its distance from the center frequency increased, with the center frequency being the highest-amplitude component. The onset phase of each stimulus varied randomly across trials, with eight possible values (0, π/4, π/2, 3π/4, π, 5π/4, 3π/2, 7π/4), with the constraint that each trial would always start with a phase different from its predecessor. All stimuli were normalized for root mean square (rms) amplitude. Within each 20-s stimulus, there were 3–5 silent gaps (average gap rate 0.21 gaps/s, sd = 0.04), each onset and offset smoothed with 3-ms half-cosine ramps. Each gap was positioned at the center of one of nine equally spaced phase bins, dividing each cycle of the frequency modulation. Gaps were separated from each other by at least 1.5 s and no gaps occurred in the first or last second of the stimulus. Note that the sounds were modulated exclusively in the frequency domain, with no rhythmic information conveyed through the amplitude parameter, which remained constant throughout each session.

**Procedure.** The experiment took place in a chamber shielded from electrical and acoustic interference, with normal lighting conditions. Participants were instructed to keep their eyes open during all tasks. Participants' sound-level thresholds were established using the method of limits. Subsequently, all stimuli were delivered at a level 55 dB higher than the individual hearing threshold (55 dB sensation level, SL).

During each session, gap duration was tailored to individual detection thresholds using an adaptive-tracking method. This method utilized two alternating staircases and a weighted up-down technique, employing custom weights as outlined in ([2]). As a result, gap durations varied between 3 and 24 ms across participants (mean = 15.13, STD = 3.7).

Prior to commencing the main experiment, participants engaged in practice trials to ensure comprehension of the task. In the main experiment, listeners were tasked with detecting gaps within the 20-s FM stimuli. They were instructed to promptly indicate gap detection by pressing a button. Participants completed five blocks, each consisting of the same set of 32 stimuli (with 4 stimuli per starting phase), arranged in different random orders. Each stimulus contained 3–5 gaps, totaling 136 gaps (ranging from 13 to 16 gaps per phase bin, with an average of approximately 15 gaps) per block. The number of phase bins was selected to achieve a balance between sampling resolution and trial count per condition, while ensuring a manageable task duration.

**tACS stimulation and data acquisition.** Behavioral data were collected in real-time using MATLAB 2017a (MathWorks) in conjunction with Psychtoolbox. The auditory stimuli, presented at a sampling rate of 44.1 kHz, were delivered through an external soundcard (RME Fireface UCX 36-channel, USB 2.0 and FireWire 400 audio interface) employing ASIO drivers. Participants listened to the stimuli via over-ear headphones (Beyerdynamic DT-770 Pro 80 Ohms, Closed-back Circumaural Dynamic with Diffuse Field Equalization, Impedance: 80 Ohm, SPL: 96 dB, Frequency range: 5–35,000 Hz). Button responses were recorded using the computer keyboard.

A battery-powered multichannel Eldith DC-stimulator Plus (NeuroConn GmbH, Ilmenau, Germany) administered tACS via two sets of circular conductive rubber electrodes affixed with electrode paste (Weaver and Company, Aurora, CO). Each electrode measured 25 mm in diameter and 2 mm in thickness. Electrodes were positioned over FC5-TP7/P7 and FC6-TP8/P8 according to the International 10–20 EEG system. It is worth noting that this electrode configuration had been previously optimized for targeting the auditory cortex in a prior study [23]. Additionally, we demonstrated in a previous study [31] that using this configuration allows for effective modulation of behavioral entrainment to FM sounds.

Each electrode pair, targeting one hemisphere, was connected to a separate stimulator channel. A third stimulator channel, employing the same waveform but at a lower current strength, was linked to an EEG amplifier (BrainAmp DC amplifiers, Brain Products GmbH) to capture the tACS signal for subsequent analysis. The tACS signals from this supplementary stimulator channel were recorded as two distinct EEG channels, utilizing a split ground.

Each participant underwent two tACS sessions. During each session, participants completed 5 blocks of the gap-detection task: 4 blocks with active tACS and 1 block with sham tACS. The active tACS was administered at the frequency of the FM stimulus (2 Hz) across 4 blocks, each lasting 11 min and 20 s. The only variation between sessions was the modulation depth of the FM stimuli, set at either 11% or 39%. The order of these conditions was randomized among participants (of the 30 participants who took part in both sessions, 17 were presented with 11% modulation depth stimuli in session 1 and 39% modulation depth in session 2, while the remaining 13 participants experienced the reverse order). The tACS current strength was fixed at 1 mA (peak-to-peak) and was gradually ramped up and down over the initial and final 10 s of each block. Due to the variability in the starting phase of the FM stimulus and the jittered inter-trial interval duration, the phase lag between the tACS signal and the FM stimulus fluctuated from trial to trial. Sham tACS was administered in a single block lasting 11 min and 20 s, with its position within the total of 5 blocks randomized across participants but kept constant across modulation depths for each participant. The sham stimulation was gradually ramped on over 10 s to simulate the sensation of stimulation onset but then ramped back down after 10 s of stimulation, resulting in a total duration of 30 s of stimulation. The stimulation waveform was sinusoidal without a DC offset, and impedance was maintained below 10 kΩ.

**Data analysis.**

**Confirming behavioral entrainment to the FM stimulus in the sham condition:** To quantify behavioral entrainment in the sham condition, hits were identified as gaps followed by a button-press response occurring no earlier than 100 ms and no later than 1,500 ms [2]. Misses were defined as gaps that either lacked a response or received a response outside of this 100–1,500 ms timeframe. Initially, for the sham condition, we computed hit rates for gap detection relative to the phase of the FM stimulus during which the gaps occurred. Subsequently, we evaluated the significance of the anticipated FM-induced modulation on gap detection behavior. For each participant and session, a cosine function was fitted to the hit rates across FM phases. The amplitude parameter of the fitted cosine function (*entAmp-FM*) served to quantify the degree of behavioral modulation by the 2 Hz FM stimulus phase. To assess the significance of this behavioral modulation (*entAmp-FM*), we employed a permutation method. This involved generating 1,000 surrogate datasets for each participant and session by randomizing the single-gap accuracy values (0, 1) while preserving their FM-stimulus phase bin labels. Cosine functions were then fitted to these surrogate datasets. Gap detection was deemed to be significantly modulated for each participant if their individual *entAmp-FM* value exceeded the 95th percentile of the distribution of *entAmp-FM* values derived from the surrogate data, corresponding to $p < 0.05$.

**Evaluating tACS effects at the group level:** To establish the tACS phase time course for each trial in the active stimulation blocks, the tACS signal recorded in the EEG underwent band-pass filtering between 1 and 10 Hz, followed by a Hilbert transform to obtain the complex output. Subsequently, the complex output was converted into phase angles using the *angle* function from MATLAB. For each gap occurrence, the phase difference between the 2-Hz FM stimulus and the 2-Hz tACS signal was determined as the angular separation between their instantaneous phases at the onset of the 2-Hz FM stimulus. Gaps were grouped based on their FM-phase (into 9 bins) and FM–tACS lag (into 6 bins, see Fig 1D), separately for each modulation depth. Hit rates were then calculated for each combination of bins. As for the sham condition, cosine fitting techniques were utilized to estimate the strength of behavioral entrainment to the FM stimulus (*entAmp-FM*) for each participant, modulation depth, and FM–tACS lag.

To assess the impact of FM–tACS lag on entrainment to the FM stimulus, we applied a second-level cosine function to model the *entAmp-FM* values as a function of FM–tACS lag. Subsequently, from this second-level fit, we identified the optimal tACS phase lag (*prefPhase-tACS*) as the FM–tACS lag that produced the highest predicted FM amplitude value. As previously documented [31], *prefPhase-tACS* values exhibited considerable variability across participants (Figs 1I and S2 Fig). To ensure an equitable assessment of tACS effects at the group level, *entAmp-FM* values were circularly adjusted to set the optimal tACS phase to zero phase for each participant and session. To mitigate potential analytical biases stemming from this alignment procedure, *entAmp-FM* values at the individual optimal tACS lag and its opposite phase were excluded from further analysis [22,31]. FM amplitude values for positive FM–tACS lags ($tACS_{(+)}$) were calculated as the average *entAmp-FM* across the two tACS lags adjacent to the re-aligned zero-lag condition, whereas FM amplitude values for negative FM–tACS lags ($tACS_{(-)}$) were calculated as the average *entAmp-FM* values across the two tACS lags adjacent to the tACS lag opposite the realigned zero-lag condition (Fig 2C–2D). Group-level tACS effects were then assessed using a repeated measures analysis of variance (rANOVA) with the two factors tACS condition (3 levels: sham, $tACS_{(+)}$, and $tACS_{(-)}$) and modulation depth (2 levels: 11 and 39%), and their interaction.

To further ensure that the observed tACS effects were not influenced by the analysis procedure, a normalization step was implemented by comparing the differences between tACS conditions (sham, $tACS_{(+)}$, and $tACS_{(-)}$) using a permutation approach [31]. For each participant and modulation depth, 1,000 surrogate datasets were generated by randomly shuffling the tACS lag assignments across trials. The same binning, realignment process, and cosine fits were then applied to each surrogate dataset, mirroring the steps taken with the original data (as described earlier). Differences between tACS conditions were assessed for both the original and surrogate datasets, with resulting values from the original data being standardized (*z*-scored) using the mean and standard deviation derived from the surrogate distributions. One-sample *t-tests* were subsequently employed to evaluate the statistical significance of these *z*-scores.

**Evaluating trial-based effects of FM-stimulus and tACS phase:** To evaluate the effect of the FM stimulus and tACS signal on a trial-by-trial basis, five mixed effects logistic regression models were tested predicting single trial detection performance from the phase of the FM stimulus (linearized using its sine and cosine), the phase of tACS (linearized using its sine and cosine), the stimulus modulation depth and their interaction (S1 Table). Models were fitted by means of the MATLAB function *fitglme* using a binomial distribution, logit link function, and Laplace fitting method. Each model included a different combination of the above-mentioned predictors. The winning model was selected as the model with the lowest Akaike's information criteria (AIC). To assess the classification significance of the top-performing model, 1,000 surrogate datasets were generated for each session. This was achieved by randomly shuffling the single-gap accuracy values (0, 1) for each participant while retaining all condition labels unchanged. Subsequently, the same mixed-effects models were fitted to these surrogate datasets. The response operator characteristic (ROC) curve and the corresponding area under the curve (AUC) were computed for both the winning model and each surrogate model. The winning model was deemed statistically significant if its AUC value exceeded the 95th percentile of the AUC distribution derived from the surrogate models ($p < 0.05$).

## Experiment 2

**Auditory stimuli.** Auditory stimuli where similar to those used in experiment 1 with the exception that the modulation depth was set to zero (Fig 3A). Gaps were still presented following the same procedure as in Experiment 1 by placing them at 9 equally spaced phase bins of an imaginary 2-Hz modulator.

**Procedure, tACS stimulation and data acquisition.** The experiment replicated the procedures outlined in Experiment 1, which encompassed the determination of individual sound levels, gap size thresholds, and training trials. The software and hardware utilized for stimulus presentation and data collection were consistent with those employed in Experiment 1. Participants engaged in a single session comprising 5 blocks, each lasting approximately 11 min and 20 s. In each block a different tACS condition was tested: sham, 0.8, 2, 3.2, and 4.4 Hz (Fig 3A). All other stimulation parameters remained unchanged from Experiment 1.

**Data analysis.** To estimate the effect of tACS on gap detection performance, gaps were first grouped by the phase of the tACS signal independently for each tACS frequency. For each tACS frequency and phase bin, hit rates were calculated as the ratio between detected gaps and all gaps presented for the given condition. For each tACS frequency, cosine functions were fitted to hit rates as a function of tACS phase. The amplitude parameter of the fitted cosine function (*entAmp-tACS*) served to quantify the extent of behavioral modulation by the tACS phase.

To assess the likelihood of observing rhythmic behavioral performance without the influence of tACS, we simulated imaginary oscillators operating at various frequencies, which were assumed to reset at the onset of the auditory stimulus in the sham condition (cosine function at a given frequency that starts with phase zero at FM-stimulus onset). Gaps were categorized based on the phase of these imaginary oscillators at gap onset, and cosine functions were applied to fit the hit rates corresponding to each oscillator phase, separately for each tACS frequency (refer to Fig 3C). The amplitude parameter of the fits were used as the baseline amplitude of rhythmic fluctuation at each frequency in the absence of tACS. To evaluate the impact of tACS at the group level, we conducted a repeated-measures ANOVA on the amplitude values obtained for each frequency from both the actual tACS stimulation and the baseline. This analysis employed a 4 × 2 design (4 frequencies: 0.8, 2, 3.2, 4.4 Hz; 2 conditions: imaginary oscillator versus tACS). As in Experiment 1, surrogate distributions were created for each participant and frequency by shuffling the gap accuracy values (1/0) across trials while keeping the tACS phase lag fixed. The *entAmp-tACS* values were *z*-scored using individual surrogate distributions, and one-sample *t*-tests were performed to evaluate the significance of the *z*-scored *entAmp-tACS* values.

To evaluate the uniformity of the distribution of *prePhase-tACS* values for each tACS frequency, the Rayleigh test was employed, as implemented in the circular statistics toolbox for MATLAB [54] (Berens, 2009).

**Oscillator model.** To explore the individual variations in optimal tACS frequency, we utilized a Stuart–Landau oscillator model with 2 ordinary differential equations and 3 parameters, as detailed in [30,55]:

$$\frac{dx}{dt} = \lambda x - \omega y - \gamma \left(x^2 + y^2\right) x + ks(t)$$

$$\frac{dy}{dt} = \lambda y + \omega x - \gamma \left(x^2 + y^2\right) y$$

In this model, *x* and *y* represent the dynamics of two coupled populations of neurons, whose interactions generate an oscillation. The frequency of this oscillation is determined by the ω parameter and its stability is controlled by λ. Positive values of λ sustain oscillations, whereas negative values lead to decay over time. The parameter γ governs the damping of the system, influencing how quickly it settles into a steady-state amplitude. The impact of tACS was simulated by altering the characteristics of the external drive *s(t)*, which was set to a sine wave with fixed frequency (set to the four tACS frequencies) and zero phase offset. The parameter *k* controls the strength of this external drive's influence on the neuronal

population, expressed as a percentage of the ongoing oscillation's amplitude. The amplitude of the ongoing oscillation was approximately 0.45 across frequencies. The model equations were numerically solved using MATLAB's *ode45* function with default parameters. Except for the intrinsic oscillator frequency ($\omega$), initial conditions and default values were consistent with reference [30]: $x = 0$, $y = -1$, $\lambda = 0.2$, and $\gamma = 1.0$. We set the tACS frequencies at 0.8, 2, 3.2, and 4.4 Hz, while systematically varying the oscillator resonance frequency ($\omega$) and tACS amplitude relative to baseline ($k$). Specifically, $k$ values ranged from 0% to 100% in increments of 10%, and $\omega$ values varied from 0.1 to 8 Hz in steps of 0.1 Hz.

For each combination of parameters, we compared the model's output with the profile of each individual obtained in Experiment 2 (illustrating *entAmp-tACS* as a function of tACS frequency). The individual's preferred frequency was determined as the oscillator resonance frequency ($\omega$) in the model simulation that demonstrated the strongest correlation with the empirical data (see Fig 4C–4D).

**Clustering analysis.** Clustering analysis of estimated individual resonance frequency was done in MATLAB using the Euclidian distance between pairs of observations and the linkage function with the method "average" to create the hierarchical tree of clusters.

## Experiment 3

**Auditory stimuli.** Auditory stimuli where similar to those used in experiment 1 with the exception that the modulation depth was of 67% and the FM rate included 0.8, 2, 3.2, and 4.4 Hz (Fig 5A), similar to the tACS frequencies applied in Experiment 2. In addition, gaps were positioned at the center of one of 15 equally spaced phase bins. Fifty-six trials were presented for each FM rate.

**Procedure.** The experiment included the same steps as in Experiment 1, including determining the individual sound level threshold, the gap size thresholds, and training trials. Software and hardware for auditory stimulation and behavioral data collection were consistent with those employed in Experiments 1 and 2. Participants engaged in a single session comprising 8 blocks, two per FM rate. The order of blocks was randomized across participants. All other stimulation parameters remained as in Experiment 1.

**EEG recording and preprocessing.** EEG recordings were performed using an actiCAP active electrode system combined with BrainAmp DC amplifiers (Brain Products GmbH). The setup included 32 Ag–AgCl electrodes mounted on a standard actiCAP 64Ch Standard-2 cap (Brain Products GmbH). Signals were continuously recorded with a passband of 0.016–1,000 Hz and digitized at a 1,000 Hz sampling rate. The reference electrode was positioned over FCz, and the ground electrode was placed over AFz. In addition to standard EEG triggers from the LPT port, stimulus markers were sent via soundcard and collected in the EEG using a Stimtrak (Brain Products GmbH). Electrode resistance was maintained below 20 k$\Omega$.

All EEG data were analyzed offline using Fieldtrip software (www.ru.nl/fcdonders/fieldtrip; version 20,240,123) and custom MATLAB scripts. The continuous raw data were segmented into 21.5 s trials, encompassing the entire FM-stimulus presentation along with an additional one-second segment before and 0.5 s after the stimulus presentation. After data segmentation, preprocessing steps included: demeaning, manual artifact rejection based on the *z*-score method, and detrending (note that outlier trials were removed before detrending). Eye blinks, muscle artifact and other noise artifacts were removed using ICA with the "runica" default method from Fieldtrip. Next, data were low-pass filtered to 30 Hz using a two-pass Butterworth filter, and trials for which the range exceeded 200 μV were removed. Last, data were re-reference to the average over all electrodes.

Preprocessed data were analyzed separately for each FM rate, with the pre- and post-stimulus time removed from each trial. Since the FM stimulus in each trial could start with one of eight different phases, trials were realigned to start at phase zero of the FM stimulus. After realignment, the data were down sampled to 500 Hz.

**Data analysis.**

**EEG frequency analysis:** To examine brain responses entrained by the FM stimulus, full-stimulus epochs were analyzed in the frequency domain. A fast Fourier transform (FFT) was performed on the trial-averaged time-domain data at each electrode after applying a Hann window. The evoked amplitude in each frequency band was calculated as the absolute value

of the complex output. Additionally, an FFT was applied to each individual trial to extract the phase angle of the complex FFT output, providing an estimate of the stimulus–brain phase lag. Intertrial phase coherence was calculated as the resultant vector length of phase angles from the complex FFT output across trials, separately for each frequency and electrode.

To estimate the strength of neural entrainment to the FM stimulus, amplitude and vector length values were extracted for each participant and averaged across a cluster of fronto-central electrodes (F3, Fz, F4, FC1, FCz, FC2, C3, Cz, C4), consistent with our previous study characterizing neural entrainment in the auditory system [2]. To control for different noise levels in estimating the amplitude and vector length at various FM frequencies (e.g., 1/$f$ noise, methodological limitations in peak estimation, and other measurement noise), the amplitude and vector length values were normalized. This was done by dividing the values observed at each frequency during the FM stimulus presentation at the same frequency by the mean amplitude recorded at the same frequency when the FM stimulus was presented at all other frequencies. Each participant's preferred FM rate for neural entrainment was identified as the FM frequency that induced the highest normalized vector length value.

**Behavioral analysis:** The strength of behavioral entrainment to the FM stimulus (*entAmp-FM*) was estimated using the same procedure as in Experiment 1. Gaps were grouped by FM frequency and stimulus phase, and cosine functions were fitted to hit rates as a function of FM phase for each FM frequency. To determine each participant's preferred FM rate for behavioral entrainment, we identified the FM frequency at which each individual exhibited the strongest amplitude of behavioral entrainment. To estimate possible differences in the optimal FM phase across FM rates, we ran multiple parametric *Hotelling* paired sample tests for equal angular means and corrected the resulting *p*-values for six comparisons using the *Bonferroni method* [54].

**Estimating the effect of FM frequency on neural and behavioral entrainment:** The effect of FM frequency on neural and behavioral entrainment was estimated by fitting linear models to the data using the MATLAB function *fitlm*. FM frequency was used as a linear regressor, while the normalized amplitude, vector length values, and *entAmp-FM* values were used as response variables, respectively.

### Surrogate distributions

To assess the significance of behavioral entrainment to the FM stimulus in Experiment 1, surrogate datasets were generated by shuffling the single gap accuracy values (1/0) across trials while keeping the FM stimulus phase fixed. To evaluate the effect of tACS on behavioral entrainment to sounds, surrogate distributions were created by shuffling the tACS phase lag relative to the sound.

In Experiment 2, the effect of each tACS frequency was assessed by comparing it to a baseline condition that estimated the strength of rhythmic perception during sham for each frequency. Additionally, to estimate the effect of tACS in the absence of a baseline oscillator, surrogate distributions were generated by shuffling single gap accuracy values (1/0) across trials while keeping the tACS phase fixed.

No surrogate distributions were used in Experiment 3, as the goal was to determine whether neural and behavioral entrainment were affected by the FM frequency, which was analyzed using a linear model. More details on these analyses are provided within each experiment.

### Other statistical analyses

For each experiment, false alarms were calculated as the number of button presses that were not preceded by a gap. Bayes factor analyses were conducted using the BayesFactor package for MATLAB. *[56]*

### Supporting information

**S1 Fig. Experiment 1.** Supplementary figure accompanying Fig 1D. Individual data showing hit rates as a function of FM-stimulus phase for each tACS condition when the FM stimulus is modulated at 11% (top) and 39% (bottom). Each plot shows data from a different participant and modulation depth. Empty plots represent participants excluded due to incomplete data. (TIF)

**S2 Fig. Experiment 1.** Supplementary figure accompanying Fig 1I. *EntAmp-FM* as a function of tACS lag for each modulation depth; 11% (top) and 39% (bottom). Solid lines show the actual amplitude parameters obtained from the initial cosine fits on the data in Fig 1D and dashed lines represent the second cosine fit to estimate the optimal tACS phase for modulating entrainment to the auditory stimulus. Each plot is a different participant and modulation depth. Empty plots represent participants excluded due to incomplete data.
(TIF)

**S3 Fig. Linking individual differences in response profiles across modalities.** The plots display entrainment amplitude as a function of frequency for Experiments 2 and 3. In Experiment 2, behavioral entrainment to tACS is quantified as (*entAmp-tACS* − baseline)/baseline, while in Experiment 3, behavioral and neural entrainment to FM sounds are represented by *entAmp-FM* and normalized vector length (VL), respectively. Data is shown for the 12 participants who took part in both experiments. Black dashed squares highlight participants with a tACS effect greater than 1 for at least one frequency. Red dashed squares indicate participants with a positive tACS effect for at least one frequency, but less than 1.
(TIF)

**S4 Fig. Effect of sample size on estimating tACS effects. (a) Histograms showing the distribution of *p*-values obtained from estimating tACS effects in 500 random samples of *N* = 27 using the data from [31].** The distributions are shown for sample of *N* = 27 using only one session data or the average across two sessions. **(b)** Probability of observing a significant tACS effect of each contrast of interest as a function of the sample size (*N* = 27–42). Note that for *N* = 42, only one independent sample can be drawn and only 42 different samples could be drawn for *N* = 41. Similar to **(a)**, the analysis was performed using data from only one session or using the average across two sessions.
(TIF)

**S1 Table. Models are organized from smallest to highest AIC.** Δ AIC relative to winning model. The winning model is also highlighted in bold. AIC, Akaike information criterion.
(DOCX)

## Acknowledgments

We thank the labs team at the Max Planck for Empirical Aesthetics for technical support and assistance with participants recruiting. We also thank Yomna Behery and Paola Najera-Maldonado for their help with data collection.

## Author contributions

**Conceptualization:** Yuranny Cabral Calderin, Molly J Henry.

**Data curation:** Yuranny Cabral Calderin.

**Formal analysis:** Yuranny Cabral Calderin.

**Funding acquisition:** Molly J Henry.

**Investigation:** Yuranny Cabral Calderin, Molly J Henry.

**Methodology:** Yuranny Cabral Calderin, Molly J Henry.

**Project administration:** Yuranny Cabral Calderin.

**Resources:** Yuranny Cabral Calderin.

**Software:** Yuranny Cabral Calderin.

**Supervision:** Yuranny Cabral Calderin, Molly J Henry.

**Validation:** Yuranny Cabral Calderin.

**Visualization:** Yuranny Cabral Calderin.

**Writing – original draft:** Yuranny Cabral Calderin.

**Writing – review & editing:** Yuranny Cabral Calderin, Molly J Henry.

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
