## [Editor Report · Decision Letter 0]

17 Sep 2024

Dear Dr Cabral Calderin, 

Thank you for submitting your manuscript entitled "Superiority of Rhythmic Auditory Signals over Electrical Stimulation to Entrain Behavior" for consideration as a Research Article by PLOS Biology.

Your manuscript has now been evaluated by the PLOS Biology editorial staff as well as by an academic editor with relevant expertise and I am writing to let you know that we would like to send your submission out for external peer review.

Once your full submission is complete, your paper will undergo a series of checks in preparation for peer review. After your manuscript has passed the checks it will be sent out for review. To provide the metadata for your submission, please Login to Editorial Manager (https://www.editorialmanager.com/pbiology) within two working days, i.e. by Sep 19 2024 11:59PM.

Kind regards,

Christian

Christian Schnell, PhD

Senior Editor

PLOS Biology

cschnell@plos.org

---

## [Decision Letter · Decision Letter 1]

22 Nov 2024

Dear Dr Cabral Calderin,

Thank you for your patience while your manuscript "Superiority of Rhythmic Auditory Signals over Electrical Stimulation to Entrain Behavior" was peer-reviewed at PLOS Biology. It has now been evaluated by the PLOS Biology editors, an Academic Editor with relevant expertise, and by several independent reviewers. 

In light of the reviews, which you will find at the end of this email, we would like to invite you to revise the work to thoroughly address the reviewers' reports.

As you will see below, the reviewers have many positive comments about your study, but they also request a few clarifications, additional experimental details and improvements of the presentation. 

Given the extent of revision needed, we cannot make a decision about publication until we have seen the revised manuscript and your response to the reviewers' comments. Your revised manuscript is likely to be sent for further evaluation by all or a subset of the reviewers.

**IMPORTANT - SUBMITTING YOUR REVISION**

*Re-submission Checklist*

*Published Peer Review*

*PLOS Data Policy*

*Blot and Gel Data Policy*

Sincerely,

Christian

Christian Schnell, PhD

Senior Editor

PLOS Biology

cschnell@plos.org

REVIEWS:

Reviewer #1 (Heather L Read): Title: "Superiority of Rhythmic Auditory Signals over Electrical Stimulation to Entrain Behavior"

EXCELLENT RATIONALE. Prior studies have shown that transcranial stimulation can modulate sensory stimulus-brain synchrony and auditory detection performance. Accordingly, the authors previously found that tACS could enhance or diminish entrainment to auditory stimulation, depending on the phase relationship between auditory and electrical signals ( Cabral-Calderin et al., 2023). Yet the auditory stimuli played a dominant role in guiding behavior on a trial-by-trial basis. This leads the team to ask how auditory and electrical entrainment of brain rhythms may interact or compete. 

HIGH SIGNIFICANCE & NOVELTY. This elegant study demonstrates that rhythmic frequency and amplitude modulated sounds are more effective at entraining slow delta oscillations than transcranial electrical stimulation (aka, tACS). Moreover, this study demonstrates that slow delta oscillations are more likely to be entrained to tACS when the frequency for stimulation is tailored to the individual's optimal entrainment frequency. A similar principle applies to the entrainment of intrinsic oscillations with natural rhythmic sensory stimulation. Thus, this study suggests a novel general principle where effective entrainment requires electric or sensory stimulus frequencies to be tailored (optimized) to the individual's intrinsic oscillatory frequencies. Tailoring entrainment with natural sensory stimuli or electrical stimuli like tACS has the potential to improve current therapeutic applications. 

APPROPRIATE computational Models & Data Analyses. 

PUBLICATION SUITABLE with minor edits. 

DATASHARING via github page is provided with MATLAB code.

MANUSCRIPT is organized and written well. 

METHODS details are sufficient for this study to be reproduced. 

DETAILED SUMMARY: 

Experiment-1. Participants listened to frequency (FM) and amplitude (AM) modulated sounds that were modulated at 2 Hz with either a small (11%) or a large (39%) amplitude modulation depth, while tACS was applied. Based on their prior study, they hypothesized that the tACS effect should be strongest for the small (11%) FM/amplitude modulated auditory stimuli. 

Result. Behavioral entrainment to the sound (entAmp-FM) was stronger for the large (39%) versus small (11%) modulation depth of sound (t (26) = 126 2.39, p = 0.024, Cohen's d = 0.63, Fig. 1e). Thus, behavioral modulation induced by rhythmic auditory stimulation varies with the sound's amplitude modulation depth. 

tACS effects did not differ across amplitude modulation depths as hypothesized. 

As in their prior study, they observed a significant boost in behavioral entrainment when tACS was optimally aligned with the stimulus rhythm. However, no significant interaction was observed across electrical and sound stimuli (F (2, 52) = 0.10), p = 0.901), suggesting that the effect of tACS was not significantly different across the amplitude modulation depths.

Experiment-2. Applied tACS at different frequencies while participants listened to unmodulated noise sounds and asked whether tACS could modulate perception when no rhythmic information was conveyed by the sound. They found that individuals varied in the optimal entrainment frequency to tACS in the absence of rhythmic sound variation, this revealed individual variability in preferred frequency for entrainment to electrical stimulation (i.e tACS). Their computational "oscillator" model further supports this idea. Prior work has shown that intrinsic synchronous oscillations are more likely to entrain to external rhythmic sensory input when the rhythm matches the individual's intrinsic neural oscillator frequency (e.g Notbohm et al., 2016). Thus, this study supports a general principle where entrainment of intrinsic neural oscillators is optimal when the frequencies are aligned to the individual's intrinsic oscillation frequency. They propose that future studies should deliver tACS frequency stimulation tailored to the individual's resonant oscillatory frequencies to potentially improve entrainment, therapeutic settings, and effects. 

Experiment-3. Collected EEG data while participants listened to FM sounds with different modulation frequencies, exploring individual variability in neural and behavioral entrainment to acoustic signals in the absence of electrical stimulation. 

Overall, acoustical rhythms were more potent in driving behavioral modulation with tACS serving as a supplementary modulator contingent on individual frequency preferences. 

MINOR EDIT SUGGESTIONS

1. Fig. 1a. The x-axis has 5 divisions over 6 seconds of sound. Would be easier to quickly understand the modulation frequency if you had 6 divisions corresponding to 1 sec each. 

2. Methods. You explain that 3-to-5 silent gaps with a mean duration of 14 (+3.7) ms were separated by > 1.5 seconds within each 20-second long sound segment. Could you please report the mean and standard deviation of the gaps rate (i.e. gaps/sec)? 

3. Lines 129-132. "During the task, hit rates were significantly higher for the 11% condition than for the 39% condition (t (26) = 3.72, p = 9.653e-04, Cohen's d = 0.75, Fig. 1g), suggesting that the task was easier for the 11% condition despite having similar gap sizes." Presumably, hit rates is the detection of gaps throughout the sound segment. In theory, the probability that the gap falls during a loud amplitude modulation is higher with the 39% amplitude modulation condition. So this is surprising. Would be good to explain a possible reason for this result. Is it attentional "competition" between the FM/AM modulation cues and the silent gap detection? 

4. Lines 370-372. "...averaged cluster of fronto-central electrodes (F3, Fz, F4, FC1, FCz, FC2, C3, Cz, and C4), in line with our previous study characterizing neural entrainment in the auditory system" Please cite your paper Cabral-Calderin & Henry J Neuro 2022 or the correct paper here. 

5. DISCUSSION, Lines 497-499. "...the mean strength of behavioral entrainment observed in Experiment 1 with modulation depths of 11% and 39% was smaller than that observed in Experiment 3 and our previous work (modulation depth of 67%, compare Figs. 1e and 5d)." Please include the citation here, given your group has multiple studies on this topic. 

6. DISCUSSION, Lines 501-504. "Previous research has shown that more strongly modulated FM signals elicit stronger neural responses [35]. This effect may occur because stronger FM signals lead to wider coverage of the frequency-tuned neuronal populations activated by the sound, due to the tonotopic organization of the auditory system. This effect may occur because stronger FM signals lead to wider coverage of the frequency-tuned neuronal populations activated by the sound, due to the tonotopic organization of the auditory system. Suggested EDIT. You could expand or revise this this to explain that columns of auditory cortical neurons that are "broadly tuned to sound frequency respond to a broader range of tonal frequencies with higher sound levels" (citations: Read HL, Winer JA, Schreiner CE. Modular organization of intrinsic connections associated with spectral tuning in cat auditory cortex. PNAS, 2001) and these same neurons respond with faster onset-offset responses to dynamic frequency and amplitude modulated sounds (Atencio CA, Schreiner CE. Spectrotemporal processing in spectral tuning modules of cat primary auditory cortex. PLoS One. 2012). Hence, they may track amplitude modulations more accurately. In theory, the FM sounds with larger amplitude modulation depths used here could more strongly activate the broadly tuned columns that follow temporal modulations well. 

7. Lines 629-634. This is an excellent point you make about how we need to explore the potential benefits of individual tailoring of tACS and auditory stimulation to optimize entrainment or other neuromodulations. 

Reviewer #2: In this paper, Cabral-Calderin and Henry, present three experiments investigating entrainment to tACS and auditory rhythms when these are presented simultaneously or in isolation. Across experiments participants were required to detect silent gaps in modulated or unmodulated noise stimuli. They found that when both stimuli were presented simultaneously, auditory stimuli dominated entrainment with no additional effect from tACS. There was evidence of entrainment when tACS was presented in isolation, but the frequency of entrainment varied across participants; similar varied entrainment frequencies were observed when sounds were presented in isolation.

Overall, the quality and rigour of the research presented is high. The manuscript is well written and the experimental protocol and data analysis presented in a clear manner. The paper will be of interest to researchers working on neurostimulation. 

I have listed comments and questions below.

1. One of the key conclusions of the paper is that when sensory and electrical stimuli compete for entrainment, sensory stimuli is more powerful. This conclusion somehow assumes that we are comparing stimuli of comparable intensities. Indeed, assuming that we are phase resetting an intrinsic oscillator, the amount of phase reset depends on the intensity of the perturbation. For both sound and tACS the intensity was kept constant, 55 dB ad 1 mA, respectively, but it is hard to know if these are 'equivalent'. While the authors do mention this as a limitation it would be relevant to reflect on this more widely, as well as the choice of stimulation intensity given current trends for higher current strengths.

2. Order of modulation depth. The authors state that each modulation depth was tested on a different session with the order randomised across participants. Is this randomised or counter-balanced? Is the order balanced across conditions?

3. Since entrainment is higher at 39% modulation, shouldn't performance also follow the same pattern? Could you add some explanation for why this is not the case?

4. The measure of accuracy used was hit rate, are the levels of false alarms or omissions equivalent between conditions?

5. Figure 1h - "each row is a different participant". But is it the same participant for each modulation depth? Please clarify in the figure legend. It would perhaps be more relevant to see the same participant for each modulation depth.

6. Figure 2d. I did not understand the continuous signal across frequencies. Please explain this analysis in more detail.

7. Figure 3d. Add more information to the legend. What is the x and y axis, what do the colours represent.

8. What are the tACS amplitude values for the oscillator model (pg 13, lines 315-316)? Do these represent tACS intensity at different phases?

9. What instructions were given to participants regarding eyes open / eyes closed?

10. Figure 6. The way in which the data is presented in this figure makes it hard to visualise how much individual participants demonstrate similar entrainment frequency preferences across modalities. Perhaps this could be addressed by adding a plot for each of the individuals that took part in experiments 2 and 3 for both modalities showing entrainment per frequency? This could be added to the supplementary. 

Reviewer #3: Yuranny Cabral-Calderin and Molly Henry present a very rich and detailed assessment of entrainment in behavioral and neural (EEG) measures, to auditory stimulation and tACS, in line with their previous work, describing three experiments. Overall, I think this is an important paper, because it sheds light on untested assumptions in the field of neural entrainment, and its careful approach yields some surprising and interesting findings, showing once more that entrainment mechanisms are more complex than initially thought. 

The methods are solidly in line with the questions asked, and the results are interpreted very carefully. 

Besides the few methodological points that I will list below, my only worry is that the paper is a bit too rich and complex, due to the multitude of findings and the complex methods used, and I therefore encourage the authors to harmonize the methods across the three experiments were possible (see suggestions below). 

Surrogate conditions across experiments: 

In Experiment 1, a bootstrapping approach was performed to obtain a surrogate condition and test whether entrainment was present or not. In Exp. 2 and 3, this seems initial condition is somewhat lifted: it is not consistently described whether significant entrainment or used as a factor in the analysis, for instance when the different entrainment frequencies are estimated (Fig. 3 b, c, d). Please include a statement about the general approach to your surrogate conditions in the methods. 

See also the first point under Experiment 2 below. 

Experiment 1: 

ll 61:

"This is important because tACS effects are often weak, and

without quantifying the relative contributions of auditory and electrical rhythms to entrainment, we cannot know how strongly tACS can push brain rhythms around while it is competing with rhythmic sensory stimulation." 

This statement seems to assume independent effects, but how about interactions, e.g. ceiling effects?

l. 128: "No significant difference in the gap size thresholds was observed between the

modulation depths."

That is indeed surprising. Please provide Bayes Factors to allow the reader to assess whether we are looking at a solid null-effect versus an inconclusive effect. 

Fig. 1 is a bit crowded, even though the panels are clear and easy to understand with the caption. Maybe move panels i - l to a new figure?

H: the realignment procedure should be described for better clarity. I do not see a semi-transparent line in panel h, and I do not understand why these values were excluded. 

Same for the description in l. 177, why would maintaining the values at the realigned peaks lead to a bias? If I understand correctly, only the phase lags were aligned, but this should not change the amplitude, correct?

Experiment 2:

The analyses described in lines 265 and follows, p. 11, test for entrainment by tACS with respect to an imaginary oscillator that is already active. 

How about the assumption that an internal oscillator is silent (in line with the Arnold-tongue approach), prior to tACS stimulation. This would mean using a different baseline, for instance phase-scrambled surrogates as in Experiment 1. Please explain why this approach was not used.

Fig. 2c: why are the amplitudes so different for the cosine fits across different tACS frequencies?

Ll 277: what is with the participants that fall outside the lower limit of the CI, aren't they noteworthy, too? 

In line with the upper point, I don't quite understand the surrogate condition here.

Exp. 2, Oscillator model, Fig. 3b: I first tried to read from the plot in 3B the tACS frequencies at which the highest entrainment amplitude was found, which seems to match the resulting resonance frequency in some, but not all cases, e.g. the left-most plot in the top row. Then I understood that the selection of the best model (individual resonance frequency) was done based on correlation of the entAmp-tACS and the model-produced values, across tACS frequencies, hence for 4 values? It would help to show these correlations / their distributions, instead of plotting entrainment amplitude on the y-axis. 

Also, the fit between the model and the data is quite striking - how do you explain that the model fits the actual amplitudes so well, for instance in participants like the fifth from the left in the bottom row, whose amplitude curve has a non-linear shape?

The analyses of individual oscillator frequencies are conducted on all participants, even though only 16 / 24 showed significant amplitude modulation by tACS (Fig. 2d). Did the results differ for those participants, and what was the reasoning to include them in the estimation of the resonance frequency?

As presented here, the estimation of the resonance frequency does not seem to have a baseline condition, which allows to assess whether there is resonance at all. In other words, a resonance frequency is always found, even if there might be no entrainment to the tACS, right? Is there a way to test if there is resonance in the first place?

Figure 3a: what is the unit of the change of entrainment, and how to interpret the y-axis?

Experiment 3:

If I understood correctly, the duration of the stimulation was constant across the different frequencies, i.e. there are more cycles in the faster condition, which might bias the measure of resultant vector length and explain the linear increase in Fig. 6c. 

l. 550, p. 21, Discussion:

The authors argue that the smaller sample size compared to their previous study might have contributed to the absence of a significant effect of entrainment in this work. It would be helpful for future studies to provide the relevant numbers (participants, trials) in the discussion paragraph. Would it be possible to compute the measures from the previous study on similar trial numbers?

The speculative point made about the similarity of the trajectories in Figure 6 an and b is not very apparent from the Figure. I would consider removing this point, to make the paper a bit less complex.

---

## [Decision Letter · Decision Letter 2]

7 Apr 2025

Dear Dr Cabral Calderin,

Thank you for your patience while we considered your revised manuscript "Superiority of Rhythmic Auditory Signals over Electrical Stimulation to Entrain Behavior" for publication as a Research Article at PLOS Biology. This revised version of your manuscript has been evaluated by the PLOS Biology editors, the Academic Editor and one of the original reviewers.

Based on the reviews and on our Academic Editor's assessment of your revision, we are likely to accept this manuscript for publication, provided you satisfactorily address the following data and other policy-related requests:

* We would like to suggest a different title to improve its accessibility for our broad audience: "Sensory stimuli are dominant over tACS electrical stimulation to modulate behavior"

* Please include the approval/license numbers of the ethical approval for your experiments.

* DATA POLICY:

Regardless of the method selected, please ensure that you provide the individual numerical values that underlie the summary data displayed in the following figure panels as they are essential for readers to assess your analysis and to reproduce it: 1EFGH, 2BDF, 3BE, 4DF, 5D and 6BCD

* CODE POLICY

We expect to receive your revised manuscript within two weeks. 

*Published Peer Review History*

*Press*

Sincerely,

Christian

Christian Schnell, PhD

Senior Editor

cschnell@plos.org

PLOS Biology

Reviewer remarks:

Reviewer #3: The authors did an excellent job in addressing the comments, and provided further analyses, and a thorough revision. 

I think this is a great paper that merits publication in PLOS Biology.

---

## [Editor Report · Decision Letter 3]

25 Apr 2025

Dear Yuranny,

Thank you for the submission of your revised Research Article "Sensory stimuli dominate over rhythmic electrical stimulation in modulating behavior" for publication in PLOS Biology. On behalf of my colleagues and the Academic Editor, Manuel Malmierca, I am pleased to say that we can in principle accept your manuscript for publication, provided you address any remaining formatting and reporting issues. These will be detailed in an email you should receive within 2-3 business days from our colleagues in the journal operations team; no action is required from you until then. Please note that we will not be able to formally accept your manuscript and schedule it for publication until you have completed any requested changes.

PRESS

Sincerely, 

Christian

Christian Schnell, PhD

Senior Editor

PLOS Biology

cschnell@plos.org